# Prosurvival long noncoding RNA *PINCR* regulates a subset of p53 targets in human colorectal cancer cells by binding to Matrin 3

Ritu Chaudhary[1], Berkley Gryder[2], Wendy S Woods[3], Murugan Subramanian[1], Matthew F Jones[1], Xiao Ling Li[1], Lisa M Jenkins[4], Svetlana A Shabalina[5], Min Mo[6], Mary Dasso[6], Yuan Yang[7], Lalage M Wakefield[7], Yuelin Zhu[8], Susan M Frier[9], Branden S Moriarity[10], Kannanganattu V Prasanth[11], Pablo Perez-Pinera[3], Ashish Lal[1]*

[1]Regulatory RNAs and Cancer Section, Genetics Branch, Center for Cancer Research, National Cancer Institute, National Institutes of Health, Bethesda, United States; [2]Oncogenomics Section, Genetics Branch, Center for Cancer Research, National Cancer Institute, National Institutes of Health, Bethesda, United States; [3]Department of Bioengineering, University of Illinois at Urbana-Champaign, Urbana, United States; [4]Laboratory of Cell Biology, Center for Cancer Research, National Cancer Institute, National Institutes of Health, Bethesda, United States; [5]National Center for Biotechnology Information, National Library of Medicine, National Institutes of Health, Bethesda, United States; [6]Laboratory of Gene Regulation and Development, National Institute of Child Health and Human Development, National Institutes of Health, Bethesda, United States; [7]Laboratory of Cancer Biology and Genetics, Center for Cancer Research, National Cancer Institute, National Institutes of Health, Bethesda, United States; [8]Molecular Genetics Section, Genetics Branch, Center for Cancer Research, National Cancer Institute, National Institutes of Health, Bethesda, United States; [9]Ionis Pharmaceuticals, Carlsbad, United States; [10]Department of Pediatrics, Masonic Cancer Center, University of Minnesota, Twin Cities, United States; [11]Department of Cell and Developmental Biology, University of Illinois at Urbana-Champaign, Urbana, United States

*For correspondence: ashish.lal@nih.gov

Competing interests: The authors declare that no competing interests exist.

**Abstract** Thousands of long noncoding RNAs (lncRNAs) have been discovered, yet the function of the vast majority remains unclear. Here, we show that a p53-regulated lncRNA which we named *PINCR* (p53-induced noncoding RNA), is induced ~100-fold after DNA damage and exerts a prosurvival function in human colorectal cancer cells (CRC) *in vitro* and tumor growth *in vivo*. Targeted deletion of *PINCR* in CRC cells significantly impaired G1 arrest and induced hypersensitivity to chemotherapeutic drugs. *PINCR* regulates the induction of a subset of p53 targets involved in G1 arrest and apoptosis, including *BTG2, RRM2B* and *GPX1*. Using a novel RNA pulldown approach that utilized endogenous S1-tagged *PINCR*, we show that *PINCR* associates with the enhancer region of these genes by binding to RNA-binding protein Matrin 3 that, in turn, associates with p53. Our findings uncover a critical prosurvival function of a p53/*PINCR*/Matrin 3 axis in response to DNA damage in CRC cells.

**eLife digest** Though DNA contains the information needed to build the proteins that keep cells alive, only 2% of the DNA in a human cell codes for proteins. The remaining 98% is referred to as non-coding DNA. The information in some of these non-coding regions can still be copied into molecules of RNA, including long molecules called lncRNAs. Little is known about what lncRNAs actually do, but growing evidence suggests that these molecules are important for a number of vital processes including cell growth and survival.

When the DNA in an animal cell gets damaged, the cell needs to decide whether to pause growth and repair the damage, or to kill itself if the harm is too great. One of the best-studied proteins guiding this decision is the p53 protein, which increases the number of protein-coding genes needed to carry out either option in this decision. That is to say that, p53 regulates the genes needed to kill the cell and the genes needed to temporarily pause its growth and repair the damage, which instead keeps the cell alive. So, how does the p53 protein guide the decision, and are lncRNA molecules involved?

Using human colon cancer cells, Chaudhary et al. now report that when DNA is damaged, the levels of a specific lncRNA increase 100-fold. Further experiments showed that this lncRNA – named *PINCR*, which refers to p53-induced noncoding RNA – promotes the survival of cells. Chaudhary et al. showed that *PINCR* molecules do this by recruiting a protein called Matrin 3 to a certain region in the DNA called an enhancer and then links it to promoter region in the DNA of specific genes that temporarily pause cell growth but keep the cell alive. This in turn activates these 'pro-survival genes'. In further experiments, when the *PINCR* molecules were essentially deleted, p53 was not able to fully activate these genes and as a result more of the cells died.

Together these findings increase our knowledge of how lncRNAs can work, especially in the context of DNA damage in cancer cells. A next important step will be to uncover other roles for the *PINCR* molecule in both cancer and healthy cells.

## Introduction

The tumor suppressor p53 functions as a sequence-specific master regulatory transcription factor that controls the expression of hundreds of genes (*Riley et al., 2008*; *Vogelstein et al., 2000*) and is mutated at a high frequency in human cancer types (*Oren, 1992*; *Vogelstein et al., 2000*; *Vousden and Lane, 2007*). Although p53 exerts its tumor suppressor effects by regulating a wide variety of cellular processes, it has context-dependent functions (*Aylon and Oren, 2016*; *Vousden, 2000*; *Zilfou and Lowe, 2009*) that are determined by various factors including cell-type, genetic background of the cell, extracellular environment, and the nature and duration of stress. Depending on the cellular context, p53 can have opposite effects on cell survival, cell migration, differentiation and metabolism (*Aylon and Oren, 2016*; *Kruiswijk et al., 2015*; *Zilfou and Lowe, 2009*).

Consistent with these pleiotropic effects of p53, the expression of genes that have opposing effects on the above-mentioned processes are regulated by p53 (*Aylon and Oren, 2016*; *Riley et al., 2008*). For example, in the context of DNA damage, p53 induces the expression of pro-survival genes such as *CDKN1A* (p21), *14-3-3σ* and *BTG2* (*Chan et al., 1999*; *Polyak et al., 1996*; *Rouault et al., 1996*) that cause cell cycle arrest, as well as proapoptotic genes such as *PUMA*, *BAX* and *NOXA* (*Riley et al., 2008*) that cause cell death. Interestingly, these prosurvival and proapoptotic genes are all upregulated by p53 in a cell regardless of the effect of p53 on cellular outcome. Therefore, it is important to investigate the function of a p53 target gene in the appropriate cellular context.

While the protein-coding genes regulated by p53 have been extensively studied and we and others have identified critical roles of microRNAs (miRNAs) in the p53 pathway (*Chang et al., 2007*; *Hermeking, 2012*; *Lal et al., 2011*; *Raver-Shapira et al., 2007*), the function of the newly discovered long noncoding RNAs (lncRNAs) in p53 signaling remains largely unknown. LncRNAs are transcripts > 200 nucleotides (nt) long that lack a functional open reading frame. Growing evidence suggests critical roles of lncRNAs in multiple cellular processes including differentiation, dosage

compensation, genomic stability, metabolism, metastasis and DNA repair (*Arun et al., 2016*; *Dey et al., 2014*; *Fatica and Bozzoni, 2014*; *Lee, 2012*; *Lee et al., 2016*; *Ling et al., 2013*; *Mueller et al., 2015*; *Redis et al., 2016*; *Sharma et al., 2015*; *Tripathi et al., 2013*). Some p53-regulated lncRNAs including *lincRNA-p21*, *PANDA*, *PINT*, *LED*, *NEAT1* and *DINO* have been shown to function as downstream effectors of p53 (*Adriaens et al., 2016*; *Blume et al., 2015*; *Dimitrova et al., 2014*; *Huarte et al., 2010*; *Hung et al., 2011*; *Léveillé et al., 2015*; *Marín-Béjar et al., 2013*; *Schmitt et al., 2016*). However, the function and mode of action of most p53-regulated lncRNAs has yet to be elucidated.

In this study, we focused on a previously uncharacterized lncRNA that we named *PINCR* (p53-induced noncoding RNA). We show that during DNA damage, *PINCR* has a context-dependent function. RNA pulldowns from cells expressing endogenous *PINCR* fused to an S1-RNA aptamer show that *PINCR* binds to the RNA-binding protein Matrin 3 to regulate the induction of a subset of prosurvival p53 targets by associating with the enhancers of these genes via a Matrin 3-p53 complex. Our results identify *PINCR* as a lncRNA that functions as a context-dependent prosurvival gene in the p53 pathway.

## Results

### Identification of *PINCR*, a p53-regulated lncRNA

To identify lncRNAs regulated by p53 in multiple cell lines, we performed microarray analysis (Affymetrix HT2.0) from three colorectal cancer (CRC) cell lines (HCT116, RKO and SW48) following activation of p53 with Nutlin-3 (*Figure 1—figure supplement 1A* and *Figure 1—figure supplement 1—source data 1*), a pharmacological inhibitor of MDM2. Using a cut-off of 1.50-fold change, 66 transcripts were upregulated in all three lines (*Figure 1—figure supplement 1B,C* and *Supplementary file 1*). Forty-eight of the 66 transcripts were also identified in a recent p53 GRO-seq study in HCT116 cells (*Allen et al., 2014*) indicating that they may be direct p53 targets. The 66 transcripts included several known p53 targets including *BTG2*, *BAX*, *CDKN1A* (p21), *GADD45A*, *MDM2* and *RRM2B*. Four out of 66 transcripts were annotated lncRNAs (*Supplementary file 2*).

Among the four lncRNAs, *RP3-326I13.1*, a ~2.2 kb long spliced intergenic lncRNA with unknown function, transcribed from the X-chromosome, was strongly induced upon p53 activation (*Supplementary file 2*). We validated this result by quantitative reverse transcription PCR (qRT–PCR) after Nutlin-3 treatment (*Figure 1A*). Due to this strong induction upon p53 activation, we named this lncRNA *PINCR*. Notably, although this lncRNA was also strongly and directly upregulated by p53 upon ectopic overexpression of p53 in a mutant p53-expressing CRC line (*Hünten et al., 2015*), its function has not been elucidated. Therefore, we decided to investigate the role of *PINCR* in the p53 network.

### *PINCR* is a nuclear lncRNA directly regulated by p53

Given the well-established role of p53 after DNA damage, we next assessed changes in *PINCR* expression during DNA damage induced by Doxorubicin (DOXO) in isogenic p53 wild-type (p53-WT) and p53 knockout (p53-KO) HCT116 and SW48 cells. The final concentration of DOXO in this and all subsequent experiments was 300 nM, unless stated otherwise. The known p53 target *PUMA* (*Nakano and Vousden, 2001*) was used as a positive control. Although *PINCR* was almost undetectable at the basal level, after DNA damage it was significantly induced as early as 8 hr after DOXO treatment and was induced >100 fold after 24 hr, in a p53-dependent manner in both lines (*Figure 1B* and *Figure 1—figure supplement 2A*).

To determine if *PINCR* is a direct target of endogenous p53, we first utilized publicly available p53 ChIP-seq (Chromatin immunoprecipitation sequencing) data (*Menendez et al., 2013*; *Nikulenkov et al., 2012*). Upon p53 activation, we observed a single p53 ChIP-seq peak in a region ~118 bp upstream of the first exon of *PINCR* in MCF7 (breast cancer) and U2OS (osteosarcoma) cells (*Figure 1C* and *Figure 1—figure supplement 2B*). We validated this result in HCT116 cells by ChIP-qPCR (*Figure 1D*). We next inserted a ~2 kb region of the *PINCR* promoter into a promoterless luciferase reporter vector (pGL3) and co-transfected this construct in HCT116 cells along with a mammalian expression vector (pCB6) or pCB6 overexpressing p53 (pCB6-p53). We found that the *PINCR* promoter drives luciferase expression upon p53 overexpression (*Figure 1E*). Deletion of

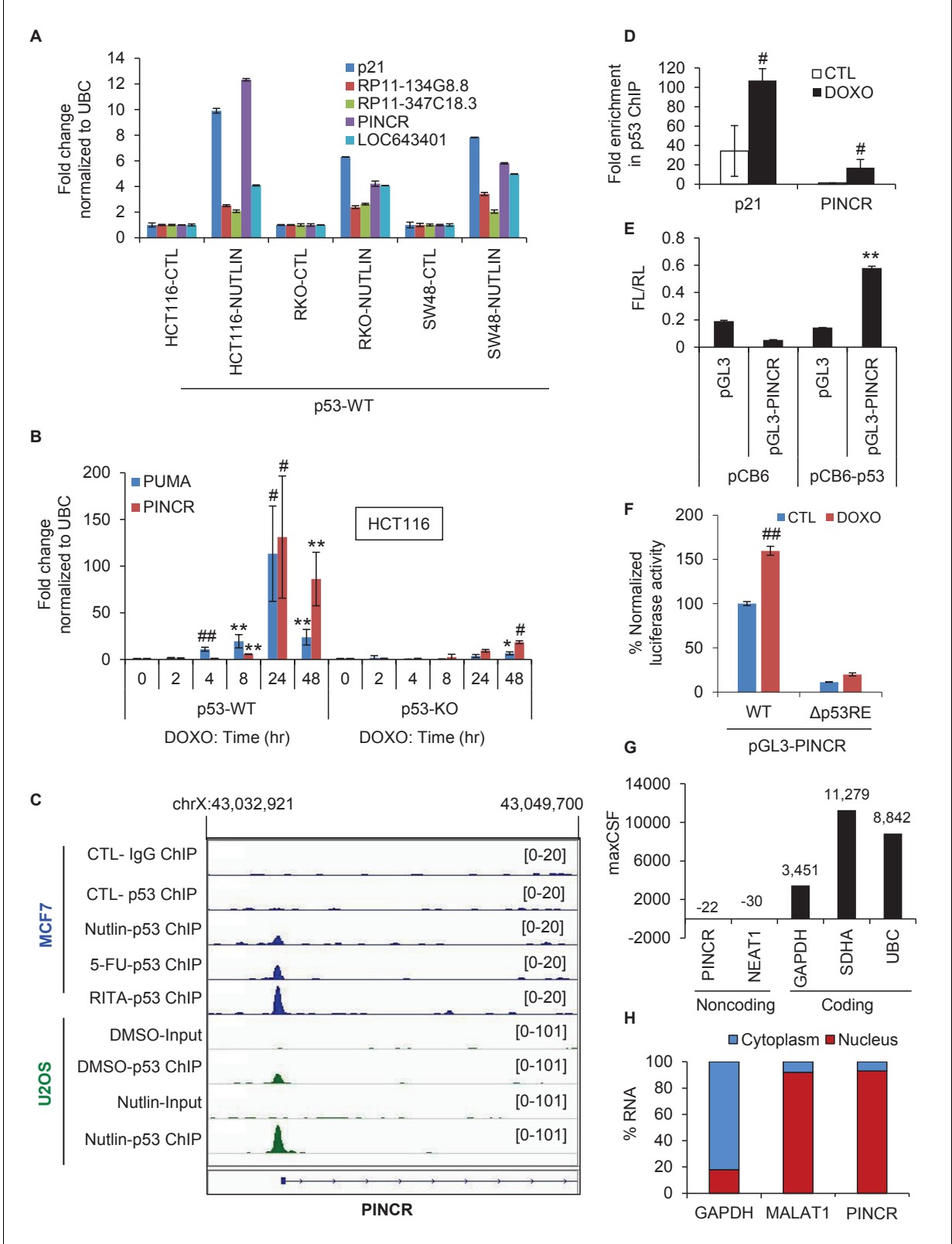

**Figure 1.** *PINCR* is a nuclear lncRNA directly induced by p53 after DNA damage. (**A**) qRT-PCR analysis from HCT116, SW48 and RKO cells untreated or treated with Nutlin-3 for 8 hr. Error bars represent SD from two independent experiments. (**B**) qRT-PCR analysis for *PINCR* and the known p53 target *PUMA* from isogenic p53-WT and p53-KO HCT116 cells untreated or treated with DOXO for the indicated times. (**C**) Snapshot of p53 ChIP-seq data of the *PINCR* promoter from MCF7 and U2OS cells untreated or treated with Nutlin or 5-FU or RITA. (**D**) HCT116 cells were untreated or treated with

*Figure 1 continued on next page*

*Figure 1 continued*

DOXO for 16 hr and qPCR using primers spanning the p53RE of *p21* and *PINCR* was performed from Input and p53-ChIP. (E) HCT116 cells were co-transfected for 48 hr with pGL3 or pGL3 containing the *PINCR* promoter, and pCB6 or pCB6-p53. Luciferase assays were performed using pRL-TK as internal control. (F) Luciferase assays were performed from untreated (CTL) or DOXO-treated HCT116 cells co-transfected for 48 hr with the internal control pRL-TK and pGL3 containing the *PINCR* wild-type (WT) promoter or pGL3 containing the *PINCR* promoter in which the p53RE was deleted (△p53RE). (G) Maximum CSF scores of *PINCR* as well as other coding and noncoding RNAs determined by analysis with PhyloCSF. (H) qRT-PCR analysis from nuclear and cytoplasmic fractions of DOXO-treated HCT116 cells; the cytoplasmic *GAPDH* mRNA and the nuclear lncRNA *MALAT1* were used as controls. Error bars in B, D-F represent SD from three independent experiments. $^{\#}p<0.01$; $^{**}p<0.005$; $^{\#\#}p<0.001$.

The following source data and figure supplements are available for figure 1:

**Figure supplement 1.** Identification of p53-regulated lncRNAs.

**Figure supplement 1—source data 1.** p53 immunoblot for *Figure 1—figure supplement 1A*.

**Figure supplement 2.** *PINCR* is highly induced after DNA damage in SW48 cells.

**Figure supplement 3.** RNA-seq was performed in duplicate from HCT116 cells untreated (CTL) or treated with DOXO (300 nM) for 16 hr (Li et al., unpublished).

**Figure supplement 4.** RT-PCR analysis of full-length *PINCR*.

**Figure supplement 5.** *PINCR* molecules per HCT116 cell.

**Figure supplement 6.** Conservation of *PINCR*.

**Figure supplement 6—source data 1.** Multiple sequence alignment of mature *PINCR* transcript for *Figure 1—figure supplement 6*.

**Figure supplement 6—source data 2.** Multiple sequence alignment of *PINCR* promoter for *Figure 1—figure supplement 6*.

the p53-response element (p53RE) in the *PINCR* promoter resulted in significant decrease in luciferase activity (*Figure 1F*). These results suggest that *PINCR* is a direct target of p53.

A detailed subsequent analysis of *PINCR* revealed many features of this lncRNA: (1) *PINCR* is a noncoding RNA because its coding potential was comparable to the noncoding RNA *NEAT1* (*Figure 1G*); (2) *PINCR* is highly enriched in the nucleus (*Figure 1H*), similar to the nuclear-retained lncRNA *MALAT1* (*Hutchinson et al., 2007*); (3) the 5'and 3'ends of *PINCR* matched the annotated transcript based on analysis of our RNA-seq data from HCT116 cells (Li et al., unpublished) (*Figure 1—figure supplement 3*); (4) analysis of the length of the *PINCR* transcript by RT-PCR revealed two closely migrating bands (*Figure 1—figure supplement 4*) that matched the expected size of the amplicon (~1.8 kb); (5) *PINCR* is expressed at ~13–26 molecules per HCT116 cell after DNA damage and less than one molecule per cell without DNA damage (*Figure 1—figure supplement 5A–C*) based on comparison of the FPKM of *PINCR* with the lncRNA *NORAD*, known to be expressed at 500–1000 molecules per HCT116 cell (*Lee et al., 2016*). As an alternative approach, qRT-PCR using *in vitro* transcribed *PINCR* RNA showed that *PINCR* is expressed at ~27 molecules per HCT116 cell after DNA damage (*Figure 1—figure supplement 5D*); (6) *PINCR* promoter including the p53RE, mature *PINCR* transcript and the transcription start site are quite conserved among primates but poorly conserved between human and mouse (*Figure 1—figure supplement 6*, *Figure 1—figure supplement 6—source data 1* and *Figure 1—figure supplement 6—source data 2*).

## Targeted deletion of *PINCR* impairs G1 arrest and results in increased cell death after DNA damage

The strong p53-dependent induction of *PINCR* after DNA damage led us to hypothesize that *PINCR* mediates the effect of p53 by regulating G1 and/or G2/M arrest after DNA damage. To begin to test this hypothesis, we used the CRISPR/Cas9 technology to delete the *PINCR* genomic locus in

HCT116 cells (*Figure 2—figure supplement 1A and B*). Targeted deletion of *PINCR* in 2 *PINCR*-KO clones (KO#1 and KO#2) was confirmed by Sanger sequencing (*Figure 2—figure supplement 1C and D*) and loss of *PINCR* expression was validated by qRT-PCR (*Figure 2A*). As negative controls, we selected two clones that were WT for *PINCR* (WT#1 and WT#2). The p53RE in the *PINCR* promoter was partially deleted in *PINCR*-KO#1 but fully intact in *PINCR*-KO#2 (*Figure 2—figure supplement 1C and D*) and as expected, we observed significantly impaired p53 binding in *PINCR*-KO#1 but not in *PINCR*-KO#2 (*Figure 2—figure supplement 2*).

We next treated the *PINCR*-WT and *PINCR*-KO cells with DOXO for 24, 48 and 72 hr and examined the effect on cell cycle arrest. In *PINCR*-KO cells, G1 arrest was substantially impaired as early as 24 hr after DNA damage (*Figure 2B*) and these cells displayed increased apoptosis as measured by the elevated sub-G1 population after 48 and 72 hr of DOXO treatment but not at 24 hr (*Figure 2C* and *Figure 2—figure supplement 3A*). Notably, loss of *PINCR* did not alter the cell cycle profile in the absence of DNA damage (*Figure 2—figure supplement 3A*).

To make sure that the observed phenotypes were not DNA-dependent but due to loss of *PINCR* RNA, we performed a rescue experiment. We inserted the full-length *PINCR* RNA into pCB6 and reintroduced *PINCR* in the *PINCR*-KO cells by stable transfection. The extent of *PINCR* overexpression was not supraphysiological; we observed ~40 fold increase in *PINCR* expression (*Figure 2D*) which is less than the ~100 fold induction that we had observed for endogenous *PINCR*. Although, reintroduction of *PINCR* in the *PINCR*-KO cells significantly rescued apoptosis at both 48 and 72 hr after DNA damage (*Figure 2E*), we did not observe a rescue of G1 arrest (*Figure 2—figure supplement 3B*). The incomplete rescue may be because unlike endogenous *PINCR* that is induced ~100 fold after DOXO-treatment, the extent of exogenous *PINCR* overexpression was ~40 fold. Another possibility is that in the rescue experiments, we overexpressed the annotated isoform, whereas we had found that HCT116 cells express at least two isoforms of *PINCR*.

In response to DOXO treatment, HCT116 cells arrest in G1 but the majority arrest in G2. p53 has been shown to play a critical role in the G1 arrest and in keeping the cells in G2 (*Bunz et al., 1998*; *Kuerbitz et al., 1992*; *Levine, 1997*). To determine if in addition to its role in G1 arrest, *PINCR* also regulates G2 arrest, we examined the integrity of the nuclear envelope by performing immunostaining for Nucleoporin after treating *PINCR*-WT and *PINCR*-KO cells with DOXO for 72 hr. We found that the nuclear membrane was intact in both *PINCR*-WT and *PINCR*-KO cells suggesting that loss of *PINCR* does not result in aberrant entry into mitosis (*Figure 2F*). Immunostaining for cleaved caspase-3, a marker of apoptosis, further confirmed increased apoptosis after DNA damage upon loss of *PINCR* (*Figure 2F* and *Figure 2—figure supplement 4A*). This hypersensitivity to DNA damage was persistent and also observed in colony formation assays (*Figure 2G* and *Figure 2—figure supplement 4B*). In this experiment, we did not observe a difference in clonogenicity upon loss of *PINCR* in untreated cells, which is consistent with the unaltered cell cycle profile upon loss of *PINCR* in untreated cells.

To confirm that *PINCR* is involved in p53-dependent G1 arrest, we performed cell cycle analysis from *PINCR*-WT and *PINCR*-KO cells after Nutlin-3 treatment. As expected, in both *PINCR*-WT and *PINCR*-KO cells, Nutlin-3 treatment resulted in dramatic reduction in the population of cells in S-phase (*Figure 2—figure supplement 5*). As compared to *PINCR*-WT cells, we observed reduced G1 population and increased G2/M population in both *PINCR*-KO clones after Nutlin-3 treatment. These data indicate that *PINCR* plays a role in p53-dependent G1 arrest and it has a prosurvival function in response to DNA damage.

## *PINCR* loss results in hypersensitivity to 5-FU and decreased tumor growth

If the major function of *PINCR* after DNA damage is to arrest cells in G1, the effect of *PINCR* loss should be more pronounced if the DNA damaging agent mainly causes G1 arrest. We therefore examined the effect on G1 arrest and apoptosis 48 hr after treatment of *PINCR*-WT and *PINCR*-KO cells with three chemotherapeutic drugs: DOXO (300 nM), the radiomimetic NCS (Neocarzinostatin, 400 ng/ml) and 5-Fluorouracil (5-FU, 100 μM). After confirming the induction of *PINCR* in response to NCS and 5-FU treatment (*Figure 3—figure supplement 1A and B*), we performed cell cycle analysis. In *PINCR*-WT cells, the percentage of cells arrested in G1 was smallest (11%) for NCS and largest (63%) for 5-FU (*Figure 3A*). Loss of *PINCR* resulted in decreased G1 arrest for NCS, DOXO and 5-FU. However, in *PINCR*-KO cells, the sub-G1 population was highest (36%) after 5-FU treatment indicating that

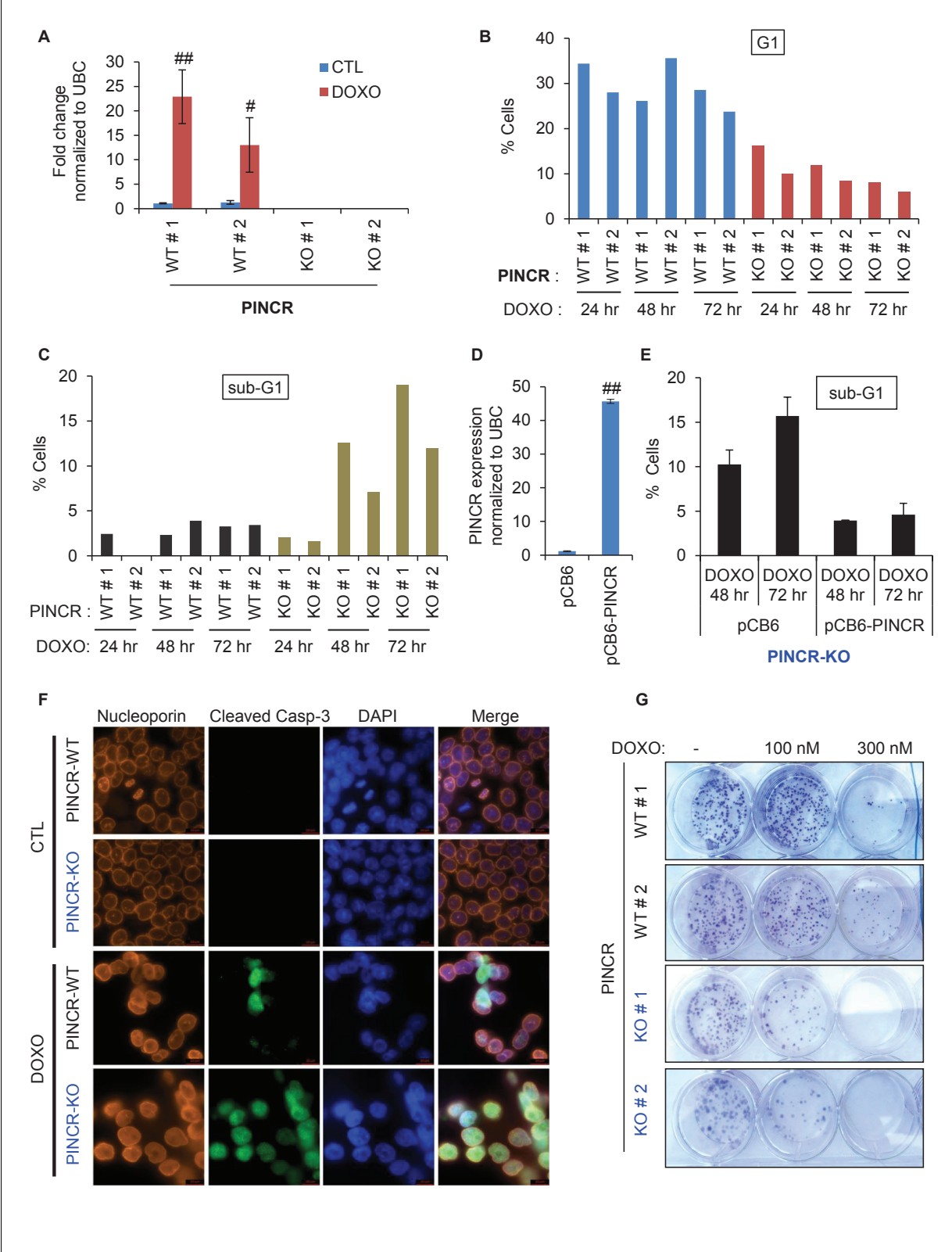

**Figure 2.** Loss of *PINCR* impairs G1 arrest and results in increased cell death after DNA damage. (**A**) qRT-PCR analysis from *PINCR*-WT (WT#1 and WT#2) and *PINCR*-KO clones (KO#1 and KO#2) untreated or treated with DOXO for 16 hr. (**B, C**) *PINCR*-WT and *PINCR*-KO clones were untreated or treated with DOXO for the indicated time points and cell cycle analysis was performed using Propidium iodide (PI) staining followed by flow cytometry analysis (FACS). (**D**) *PINCR*-KO cells were stably transfected with pCB6 or pCB6-*PINCR* and qRT-PCR was performed. (**E**) *PINCR*-KO cells stably
*Figure 2 continued on next page*

*Figure 2 continued*

expressing *PINCR* were untreated or treated with DOXO in biological duplicates at the indicated times and cell death (sub-G1 cells) was assessed by PI staining followed by FACS. (F) Immunostaining for Nucleoporin and cleaved caspase-3 from *PINCR*-WT and *PINCR*-KO clones with or without DOXO treatment for 72 hr. DNA was counterstained with DAPI. (G) *PINCR*-WT and *PINCR*-KO clones were untreated or treated with the indicated DOXO concentrations for 4 hr and colony formation assays were performed after 10 days. Error bars in A and D represent SD from three independent experiments. #$p<0.01$; ##$p<0.001$.

The following figure supplements are available for figure 2:

**Figure supplement 1.** CRISPR deletion of *PINCR* locus.

**Figure supplement 2.** p53 binding to the p53RE of *PINCR* in *PINCR*-WT and *PINCR*-KO cells was assessed by ChIP-qPCR from HCT116 cells (*PINCR*-WT or *PINCR*-KO) treated with 5-FU for 24 hr.

**Figure supplement 3.** Cell cycle profiles for *Figure 2*.

**Figure supplement 4.** Quantitation for immunostaining and colony formation.

**Figure supplement 5.** Loss of *PINCR* results in reduced G1 arrest after Nutlin-3 treatment.

hypersensitivity of *PINCR*-KO cells to chemotherapeutic drugs is dependent on G1 arrest. Importantly, this impaired G1 arrest and increased apoptosis after 5-FU treatment upon loss of *PINCR* was observed in both *PINCR*-KO clones (*Figure 3B and C*) and was further confirmed by immunoblotting for the apoptosis marker cleaved-PARP (*Figure 3D* and *Figure 3—source data 1*). Furthermore, loss of *PINCR* significantly impaired clonogenicity after 5-FU treatment (*Figure 3E and F*).

Next, to determine if the observed phenotypes are not restricted to HCT116, we knocked out *PINCR* in SW48 cells (*Figure 3—figure supplement 2A and B*). In response to DNA damage induced by 5-FU, we observed reduced G1 arrest and increased apoptosis in the *PINCR*-KO clone as compared to *PINCR*-WT clones (*Figure 3—figure supplement 3A–C*). Moreover, following extended treatment with 5-FU, the *PINCR*-KO clone was markedly more sensitive than *PINCR*-WT clones (*Figure 3—figure supplement 4*). These data confirm that the phenotypic effects observed upon loss of *PINCR* are not unique to HCT116.

We next employed several different concentrations of 5-FU (0 to 375 μM) and measured the extent of induction of *PINCR* and *PUMA*, and examined the sub-G1 population. Although we found an increase in sub-G1 population with increasing dose of 5-FU, the extent of induction of *PINCR* or *PUMA* did not change significantly (*Figure 3—figure supplement 5A*). At all doses of 5-FU, *PINCR*-KO cells were more sensitive than *PINCR*-WT cells (*Figure 3—figure supplement 5B*). These data indicate that the extent of *PINCR* induction may not be determinant of the likelihood of the cells to die rather than undergo G1 arrest. In addition, we found that over-expression of *PINCR* did not significantly affect the cell cycle in untreated cells or in response to DNA damage induced by 5-FU (*Figure 3—figure supplement 6*).

To determine the function of *PINCR* in an *in vivo* setting, we subcutaneously injected NOD-SCID mice with HCT116-*PINCR*-WT or *PINCR*-KO cells, untreated or treated with 5-FU for 4 hr followed by a 4-hr recovery. Although mice injected with 5-FU-treated *PINCR*-WT or *PINCR*-KO cells did not form tumors, in untreated condition the rate of tumor growth was substantially reduced (7–10-fold) upon loss of *PINCR* (*Figure 3G and H*). All mice injected with untreated *PINCR*-WT cells developed detectable tumors, whereas the untreated *PINCR*-KO cells displayed significantly reduced tumor growth as early as day 12 post-injection ($p<0.05$) (*Figure 3G*). Immunohistochemical staining of the tumors for the proliferation marker Ki67 and the apoptosis marker cleaved caspase-3 revealed that *PINCR*-WT and *PINCR*-KO tumors had a high proportion of Ki67-positive cells (>50%) and a very low proportion of cleaved caspase-3-positive cells (<1%) (*Figure 3—figure supplement 7*). As compared to *PINCR*-WT tumors, the *PINCR*-KO tumors had significantly decreased Ki67-positive cells (*Figure 3—figure supplement 7A* and *Figure 3—figure supplement 7—source data 1*), suggesting that the observed reduced tumor volume is due to inhibition of cell proliferation.

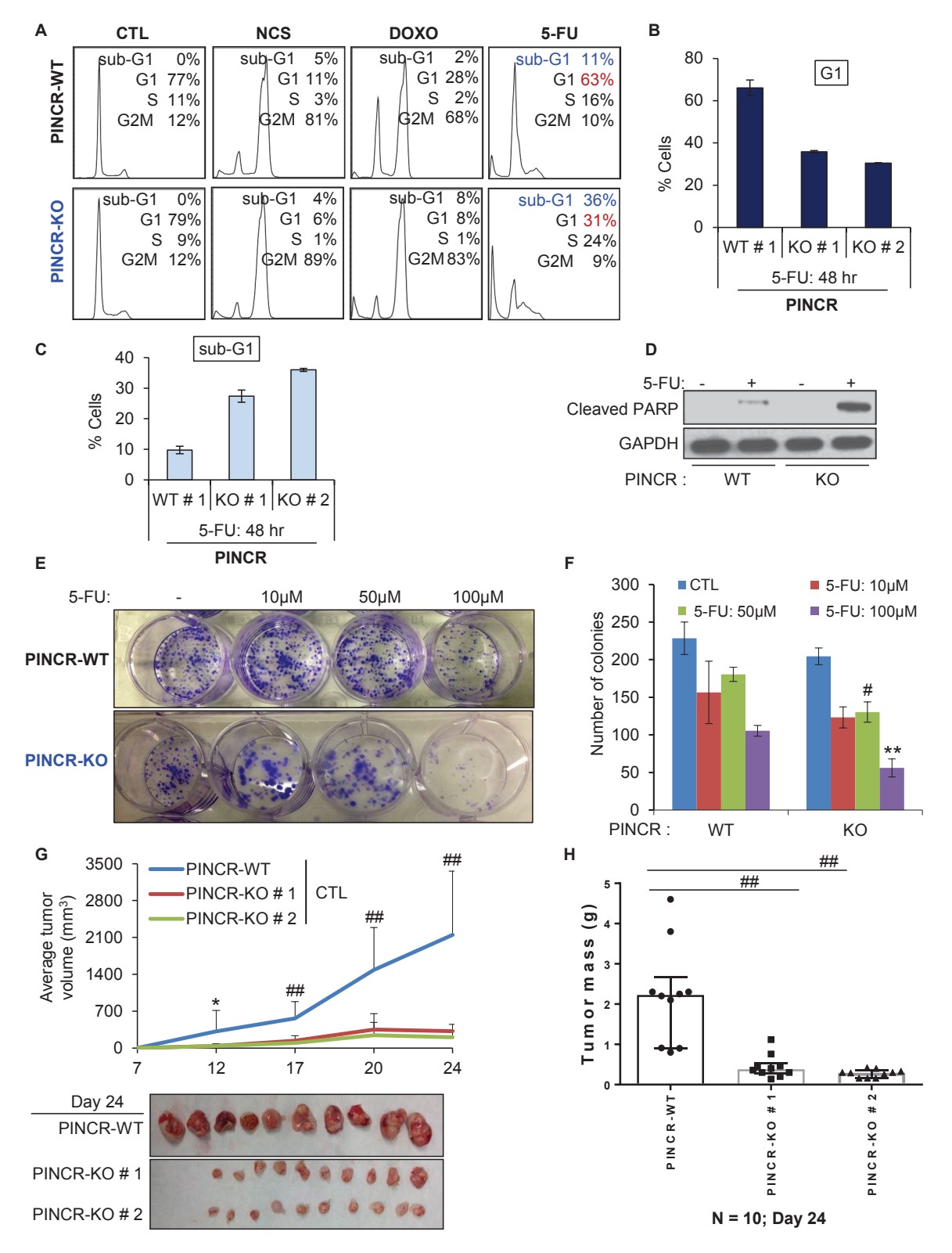

**Figure 3.** *PINCR* knockout cells are hypersensitive to 5-FU *in vitro* and poorly tumorigenic *in vivo*. (A) *PINCR*-WT and *PINCR*-KO clones were untreated or treated with NCS, DOXO, 5-FU for 48 hr and PI staining followed by FACS analysis was performed. (B, C) *PINCR*-WT#1 and *PINCR*-KO (KO#1 and KO#2) clones were untreated or treated with 5-FU in biological duplicates for 48 hr and the effect on G1 arrest and cell death (sub-G1) was assessed by PI staining followed by FACS. (D) *PINCR*-WT and *PINCR*-KO cells were untreated or treated with 5-FU for 48 hr and immunoblotting for cleaved PARP

*Figure 3 continued on next page*

*Figure 3 continued*

and loading control GAPDH was performed. (E, F) *PINCR*-WT and *PINCR*-KO cells were untreated or treated with indicated 5-FU concentrations for 4 hr, and colony formation assays were performed after 10 days. (G, H) Untreated *PINCR*-WT#1 and *PINCR*-KO (KO#1 and KO#2) cells were injected subcutaneously into the flanks of athymic nude mice (five mice for each group, two tumors per mice). Average tumor volume (G) and tumor mass (H) are shown. Error bars in F represent SD from three experiments. Tumor mass data in H is shown as median +/- interquartile range and p-values were calculated using the Krusal-Wallis test. $*p<0.05$; $\#p<0.01$; $**p<0.005$; $\#\#p<0.001$.

The following source data and figure supplements are available for figure 3:

**Source data 1.** Cleaved PARP immunoblot for *Figure 3D*.

**Figure supplement 1.** *PINCR* is induced by 5-FU or NCS.

**Figure supplement 2.** CRISPR knockout of *PINCR* in SW48 cells.

**Figure supplement 3.** Cell cycle profiles for *PINCR*-WT and *PINCR*-KO SW48 cells.

**Figure supplement 4.** Long term proliferation for *PINCR*-WT and *PINCR*-KO SW48 cells.

**Figure supplement 5.** Effect of different doses of 5-FU on *PINCR* levels and cell survival.

**Figure supplement 6.** Over-expression of *PINCR* has no significant effect on cell cycle.

**Figure supplement 7.** Immunohistochemical analysis from *PINCR*-WT and *PINCR*-KO cells.

**Figure supplement 7—source data 1.** Ki67 staining images of *PINCR*-WT and *PINCR*-KO tumors for *Figure 3—figure supplement 7A*.

## *PINCR* regulates the induction of a subset of p53 targets after DNA damage

To determine if *PINCR* mediates its effect by regulating gene expression, we performed mRNA microarrays from three biological replicates of *PINCR*-WT and *PINCR*-KO cells, untreated or treated with 5-FU for 24 hr (*Supplementary file 3*). Gene set enrichment analysis (GSEA) for the upregulated genes identified the p53 pathway as the top upregulated pathway after DNA damage in both *PINCR*-WT and *PINCR*-KO cells (*Figure 4—figure supplement 1A* and *Figure 4—figure supplement 1—source data 1*) suggesting that loss of *PINCR* does not alter global p53 signaling. Consistent with this, we observed comparable p53 induction in both *PINCR*-WT and *PINCR*-KO cells after 5-FU treatment (*Figure 4—figure supplement 1B*) and the majority of known p53 targets including the G1 regulator *p21* were induced to similar levels. Interestingly, the normalized enrichment score (NES) for the p53 pathway in *PINCR*-KO cells was significantly lower (NES = 2.673) than that in *PINCR*-WT cells (NES = 3.045) indicating that the induction of a subset of p53 targets may be abrogated in *PINCR*-KO cells (*Figure 4—figure supplement 1A*). Thus, the induction of a subset of p53 targets appeared to be *PINCR*-dependent (*Figure 4A*). Further analysis indicated that the induction of 11 direct p53 targets that were also identified in the p53 GRO-seq study (*Allen et al., 2014*) including *BTG2*, *GPX1*, *RRM2B* was less pronounced in *PINCR*-KO cells (*Supplementary file 3*).

Among the 11 *PINCR*-dependent p53 targets, we selected *BTG2*, *RRM2B* and *GPX1* for further analysis due to evidence in the literature supporting their roles in induction of G1 arrest and inhibition of apoptosis after DNA damage. *BTG2* encodes an antiproliferative protein critical in regulation of the G1/S transition (*Guardavaccaro et al., 2000*; *Rouault et al., 1996*; *Tirone, 2001*). Silencing *RRM2B* in p53-proficient cells reduces ribonucleotide reductase activity, DNA repair, and cell survival after exposure to various genotoxins (*Tanaka et al., 2000*; *Xue et al., 2007*; *Yanamoto et al., 2005*). *GPX1* attenuates DOXO-induced cell cycle arrest and apoptosis (*Gao et al., 2008*).

In subsequent experiments, we sought to use *p21* as a negative control because *p21* mRNA was induced to similar levels in both *PINCR*-WT and *PINCR*-KO cells (*Figure 4B*). However, given the well-established role of p21 in controlling G1 arrest after DNA damage, it was important to make sure that loss of *PINCR* did not alter p21 protein levels, p21 subcellular localization and/or Rb-

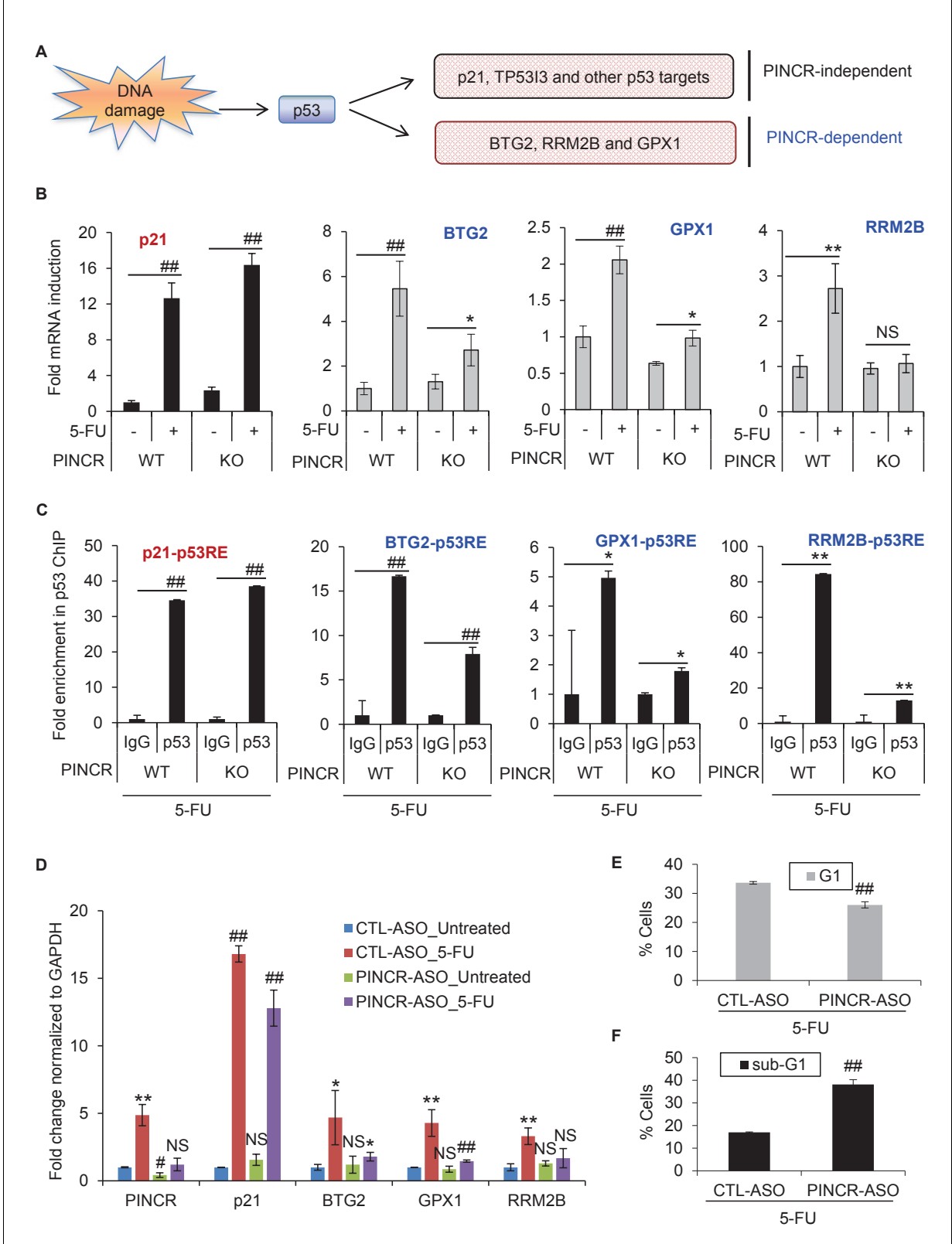

**Figure 4.** *PINCR* regulates the induction of select p53 target genes important for G1 arrest after DNA damage. (**A**) Schematic representation of a subset of p53 target genes upregulated after 5-FU treatment in a *PINCR*-dependent or *PINCR*-independent manner. (**B**) qRT-PCR analysis from *PINCR*-WT and *PINCR*-KO cells untreated or treated with 5-FU for 24 hr. (**C**) *PINCR*-WT and *PINCR*-KO cells were treated with 5-FU for 24 hr and qPCR for the p53RE of *BTG2, RRM2B* and *GPX1* was performed from IgG-ChIP and p53-ChIP. (**D–F**) HCT116 cells were reverse transfected with CTL-ASO or *PINCR-*
*Figure 4 continued on next page*

*Figure 4 continued*

ASO for 48 hr. The cells were then left untreated or treated with 5-FU for 24 hr (**D**) or 48 hr (**E, F**) following which qRT-PCR analysis (**D**), PI staining and FACS analysis was performed (**E, F**). Error bars in **B-F** represent SD from three independent experiments. *$p < 0.05$; #$p < 0.01$; **$p < 0.005$; ##$p < 0.001$.

The following source data and figure supplements are available for figure 4:

**Figure supplement 1.** Loss of *PINCR* results in impaired induction of a subset of p53 targets without altering induction of p53 levels.

**Figure supplement 1—source data 1.** p53 immunoblot for *Figure 4—figure supplement 1B*.

**Figure supplement 2.** Loss of *PINCR* does not markedly alter total p21 protein levels, Rb phosphorylation or subcellular localization of p21.

**Figure supplement 2—source data 1.** p21 and phospho Rb immunoblots for *Figure 4—figure supplement 2A, B and C*.

**Figure supplement 3.** Knockdown of the *PINCR* targets *BTG2, GPX1* or *RRM2B* phenocopies the effect of *PINCR* loss.

**Figure supplement 4.** Raw cell cycle profiles showing that knockdown of the *PINCR* targets *BTG2, GPX1* or *RRM2B* phenocopies the effect of *PINCR* loss.

**Figure supplement 5.** Knockdown of *PINCR* results in decreased G1 arrest and increased apoptosis.

**Figure supplement 6.** Knockdown of *PINCR* results in reduced colony formation.

phosphorylation. Indeed, we found similar levels of total, nuclear or cytoplasmic p21 in *PINCR*-WT and *PINCR*-KO cells under untreated condition and after 5-FU treatment (*Figure 4—figure supplement 2A and B* and *Figure 4—figure supplement 2—source data 1*). The decrease in Rb phosphorylation in response to 5-FU treatment was comparable in *PINCR*-WT and *PINCR*-KO cells (*Figure 4—figure supplement 2C* and *Figure 4—figure supplement 2—source data 1*). These results indicate that p21 expression is not altered upon loss of *PINCR* and it can therefore be used as a negative control.

We next asked the question if depletion of the *PINCR* targets *BTG2, GPX1* and *RRM2B* recapitulated the effects of *PINCR* depletion. We validated significant knockdown of these genes by qRT-PCR (*Figure 4—figure supplement 3A*) and found significantly increased apoptosis (sub-G1 cells) upon knockdown of each of these genes followed by 5-FU treatment (*Figure 4—figure supplement 3B–3D* and *Figure 4—figure supplement 4*). Significant reduction in G1 arrest after 5-FU treatment was observed after knockdown of *GPX1* but not *BTG2* or *RRM2B*. These data indicate that depletion of *BTG2, GPX1* or *RRM2B* recapitulates the effects of *PINCR* depletion in response to 5-FU treatment.

Consistent with our microarray data, we observed impaired induction of *BTG2, GPX1* and *RRM2B* mRNAs upon loss of *PINCR* (*Figure 4B*). Moreover, by p53 ChIP-qPCR, we observed substantial decrease in the binding of p53 to the p53RE of *BTG2, GPX1* and *RRM2B* upon loss of *PINCR* (*Figure 4C*). There is evidence in the literature that p53 can directly bind to RNA including a recent report showing direct binding of p53 to the p53-regulated lncRNA *DINO* (*Riley and Maher, 2007*; *Schmitt et al., 2016*). However, we found that *PINCR* does not directly bind to p53 (data not shown).

To make sure that the altered induction of *BTG2, GPX1* and *RRM2B* reflect a function of the *PINCR* transcript itself, we measured the induction of these genes after *PINCR* knockdown using antisense oligonucleotides (ASOs). We tested 5 ASOs that potentially target *PINCR* RNA (data not shown). Robust knockdown of *PINCR* in HCT116 cells was observed with one ASO that we designated as *PINCR*-ASO (*Figure 4D*). Importantly, as observed with *PINCR*-KO cells, knockdown of *PINCR* followed by 5-FU treatment resulted in decreased induction of *BTG2, GPX1* and *RRM2B* but not *p21* (*Figure 4D*) and caused decreased G1 arrest (*Figure 4E*) and increased apoptosis (*Figure 4F*) after 5-FU or DOXO treatment (*Figure 4—figure supplement 5*). In clonogenic survival assays, knockdown of *PINCR* resulted in reduced colony formation after 5-FU treatment (*Figure 4—figure supplement 6A*). Surprisingly, unlike *PINCR*-KO cells that did not show significant difference

in proliferation from *PINCR*-WT cells in untreated condition, decreased colony formation in untreated condition was observed after *PINCR* knockdown (*Figure 4—figure supplement 6B*). Although this result indicates that basal *PINCR* levels can regulate proliferation despite low expression, this growth defect may be restored long-term during genetic deletion of *PINCR* using CRISPR/Cas9. Taken together, the results from *PINCR* knockdown experiments corroborates our findings from the *PINCR*-KO clones.

## RNA pulldowns and mass spectrometry identifies Matrin 3 as a *PINCR*-interacting protein that mediates the effect of *PINCR*

Because we found that *PINCR* does not bind to p53, we hypothesized that *PINCR* binds to an RNA-binding protein that serves as an adaptor protein and mediates this effect of *PINCR*. To identify this adaptor protein, we incubated *in vitro*-transcribed biotinylated (Bi)-*PINCR* (Bi-*PINCR*) or Bi-*Luciferase* (Bi-*LUC*) RNA with untreated or DOXO-treated nuclear extracts and performed streptavidin pulldowns followed by mass spectrometry. Eleven proteins were enriched at least twofold in the Bi-*PINCR* pulldowns (*Supplementary file 4*) as compared to Bi-*LUC* pulldowns in untreated condition as well as after DOXO treatment. Of these 11 proteins, the RNA- and DNA-binding nuclear matrix protein Matrin 3 showed the strongest enrichment (eightfold in untreated condition; 16-fold after DOXO treatment) (*Figure 5A* and *Supplementary file 4*). In a recent iCLIP (Individual-nucleotide resolution UV crosslinking and immunoprecipitation) study (*Coelho et al., 2015*), the consensus RNA motif recognized by Matrin 3 was identified. We found that *PINCR* has six Matrin 3 binding motifs, and this motif was significantly enriched in the *PINCR* RNA as compared to the transcriptome (*Figure 5—figure supplement 1A and B*). We next validated the specific *PINCR*-Matrin 3 interaction by performing streptavidin pulldowns followed by immunoblotting after incubating Bi-*PINCR* or Bi-*LUC* with HCT116 nuclear extracts (*Figure 5B* and *Figure 5—source data 1*) or recombinant Matrin 3 (rMatrin 3) (*Figure 5C* and *Figure 5—source data 1*). Moreover, we observed ~300-fold enrichment of *PINCR* in the Matrin 3 IPs from formaldehyde crosslinked HCT116 cells treated with DOXO (*Figure 5D*); *p21* mRNA was not enriched (*Figure 5D* and *Figure 5—source data 1*), demonstrating the specificity of the *PINCR*-Matrin 3 interaction.

Next, we knocked down Matrin 3 with two independent siRNAs (*Figure 5E* and *Figure 5—figure supplement 1C*) and determined the effect on G1 arrest and apoptosis of *PINCR*-WT and *PINCR*-KO. After 5-FU or DOXO treatment, there was more apoptosis upon Matrin 3 knockdown in *PINCR*-WT but this increase was not observed in the *PINCR*-KO (*Figure 5F* and *Figure 5—figure supplements 2* and *3*) indicating that Matrin 3 is a downstream effector of *PINCR*. We did not observe a significant difference in G1 arrest, suggesting that Matrin 3 does not mediate the G1 arrest regulated by *PINCR*. These data indicate that there is an epistatic interaction between *PINCR* and Matrin 3 as virtually all the apoptotic effects of Matrin 3 after DNA damage are dependent on *PINCR*. Furthermore, in *PINCR*-WT cells silencing Matrin 3 resulted in less or no induction of the *PINCR* targets *BTG2*, *GPX1* and *RRM2B* but not *p21* mRNA after 5-FU treatment (*Figure 5G*). A role of Matrin 3 in regulating the induction of these p53 targets was also observed in response to Nutlin-3 treatment (*Figure 5—figure supplement 4*). Immunoblotting from untreated or 5-FU-treated HCT116 whole cell lysates and nuclear and cytoplasmic lysates indicated no change in Matrin 3 levels or subcellular localization (*Figure 5—figure supplement 5* and *Figure 5—figure supplement 5—source data 1*). Collectively, these data suggest that the induction of the *PINCR* targets *BTG2*, *GPX1* and *RRM2B* is largely mediated by Matrin 3 and reveal an epistatic interaction between *PINCR* and Matrin 3.

## Matrin 3 interacts with p53 and associates with the p53RE of select *PINCR* targets

To determine if Matrin 3 mediates the effect of *PINCR* by functioning as an adaptor protein, we performed co-IP experiments to determine if p53 and Matrin 3 form a complex. We found that p53 interacts with Matrin 3 in both untreated and 5-FU treated cells (*Figure 6A and B*, *Figure 6—figure supplement 1A*, *Figure 6—source data 1* and *Figure 6—figure supplement 1—source data 1*). This interaction was not altered in the presence of RNase A or DNase, suggesting that p53 and Matrin 3 form a protein-protein complex (*Figure 6B*). This result prompted us to examine the association of Matrin 3 on the p53RE of *BTG2*, *GPX1* and *RRM2B*. Matrin 3 ChIP-qPCR revealed that in untreated *PINCR*-WT cells, Matrin 3 binds to the p53RE of *BTG2* and *RRM2B* but not *GPX1*

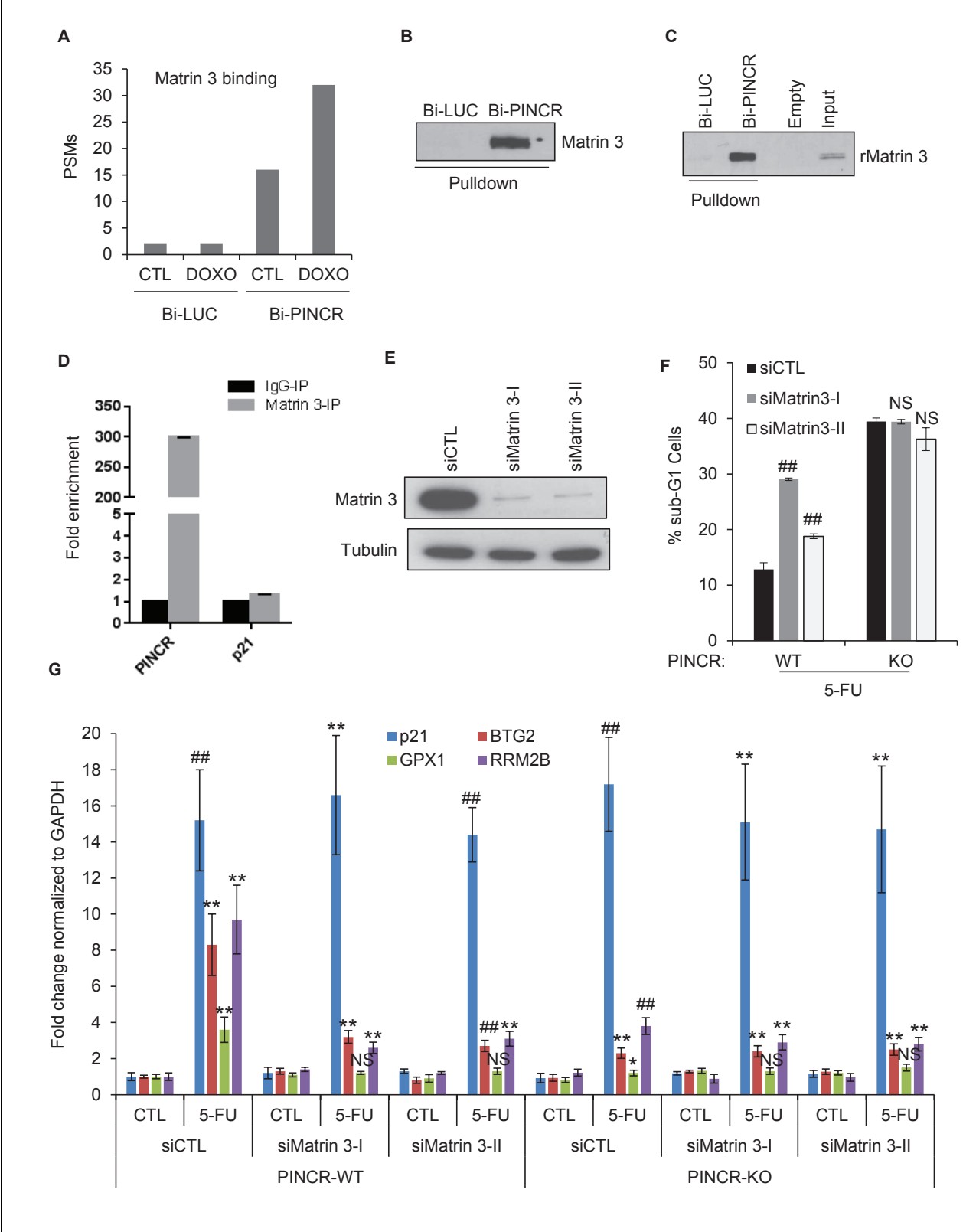

**Figure 5.** Matrin 3 binds to *PINCR* and functions as a downstream effector of *PINCR*. (**A**) Peptide spectrum matches (PSMs) corresponding to Matrin 3 in the Bi-*LUC* and Bi-*PINCR* pulldowns from mass spectrometry analysis. (**B, C**) Streptavidin pulldowns followed by immunoblotting was performed following incubation of Bi-*LUC* and Bi-*PINCR* RNA with DOXO-treated HCT116 nuclear extracts (**B**) or recombinant Matrin 3 (rMatrin 3) (**C**). (**D**) Specific enrichment of *PINCR* in the Matrin 3 IPs was assessed by qRT-PCR from 24 hr 5-FU-treated formaldehyde cross-linked HCT116 cells. *p21* mRNA was

*Figure 5 continued on next page*

*Figure 5 continued*

used as negative control. (**E**) *PINCR*-WT cells were transfected with CTL or two independent Matrin 3 siRNAs (I and II) for 48 hr and Matrin 3 knockdown was measured by immunoblotting. (**F**) *PINCR*-WT and *PINCR*-KO cells were transfected with CTL or Matrin 3 siRNAs and after 48 hr the cells were untreated or treated with 5-FU for 48 hr. The effect on the sub-G1 population was assessed by PI staining followed by FACS. (**G**) *PINCR*-WT and *PINCR*-KO cells were transfected with CTL or Matrin 3 siRNAs for 48 hr; transfected cells were left untreated or treated with 5-FU for 24 hr and qRT-PCR was performed. Error bars in D, F and G represent SD from three independent experiments. *p<0.05; #p<0.01; **p<0.005.
The following source data and figure supplements are available for figure 5:

**Source data 1.** Matrin 3 immunoblot for *Figure 5B, C and E*.
**Figure supplement 1.** Matrin 3 motifs in *PINCR* RNA.
**Figure supplement 2.** Cell cycle analysis after Matrin 3 knockdown.
**Figure supplement 3.** Raw cell cycle profiles from a representative experiment for *Figure 5F* and *Figure 5—figure supplement 3* are shown.
**Figure supplement 4.** Matrin 3 regulates the induction of *PINCR* targets upon p53 activation by Nutlin-3.
**Figure supplement 5.** Matrin 3 protein level and subcellular localization is not altered after DNA damage.
**Figure supplement 5—source data 1.** Matrin 3 immunoblot for *Figure 5—figure supplement 5A and B*.

(*Figure 6C*). For all three genes, in *PINCR*-WT cells, there was increased Matrin 3 binding to their p53RE after 5-FU treatment. Loss of *PINCR* impaired this binding of Matrin 3 to the p53RE of these genes (*Figure 6C*) but not the p53RE of *p21* (*Figure 6—figure supplement 1B*). Interestingly, after knockdown of Matrin 3 and 5-FU treatment in *PINCR*-WT cells, we did not observe a significant difference in the binding of p53 to the p53RE of these genes (*Figure 6D* and *Figure 6—figure supplement 2*) suggesting that Matrin 3 does not control p53 occupancy at the p53RE in the promoters of these genes.

## Matrin 3 associates with enhancers within insulated neighborhoods of *PINCR* targets

We next examined how Matrin 3 regulates the induction of the *PINCR* targets *BTG2*, *GPX1* and *RRM2B* after DNA damage, without altering p53 binding to their promoters. Given the evidence that Matrin 3 associates with enhancer regions (*Skowronska-Krawczyk et al., 2005*), we reasoned that Matrin 3 modulates the induction of these genes by binding to their enhancer regions. More recently, it has been shown that proper enhancer-gene pairing is enabled by insulated neighborhoods formed by CTCF anchoring at domain boundaries and cohesion looping (*Hnisz et al., 2016*) and are mostly conserved across cell types. To test this possibility, we identified insulated neighborhoods around *PINCR* targets, ChIP-seq data tracks (*ENCODE Project Consortium, 2012*) in HCT116 cells (*Figure 7A*, *Figure 7—figure supplement 1* and *Figure 7—figure supplement 2*) for (1) CTCF, a protein known to bind to chromatin domain boundaries, (2) the chromatin loop-enabling cohesion component RAD21, (3) promoter associated histone mark H3K4me3 and (4) active promoter/enhancer associated mark H3K27ac. To determine the overlap of these peaks with p53, we utilized p53 ChIP-seq data from MCF7 and U2OS cells (*Figure 7A*, *Figure 7—figure supplement 1* and *Figure 7—figure supplement 2*). In addition, to determine potential chromatin looping near these *PINCR* targets, we utilized Hi-C data (*Figure 7A*, *Figure 7—figure supplement 1* and *Figure 7—figure supplement 2*). For each of the three *PINCR* targets, the Hi-C data indicated chromatin looping with appropriate CTCF and cohesion signal consistent with insulated domain structure. Within the loop, we observed the following: (1) a strong p53 ChIP-seq peak corresponding to the p53RE in the promoter of these genes that was also marked by strong signal for H3K4me3 and H3K27ac; (2) a weak p53 ChIP-seq peak that was also marked by strong signal for H3K27ac and weak signal H3K4me3. Promoters are marked by high H3K4me3 and high H3K27ac whereas enhancers typically have low H3K4me3 and high H3K27ac (*Ernst et al., 2011*). Thus, our Hi-C and

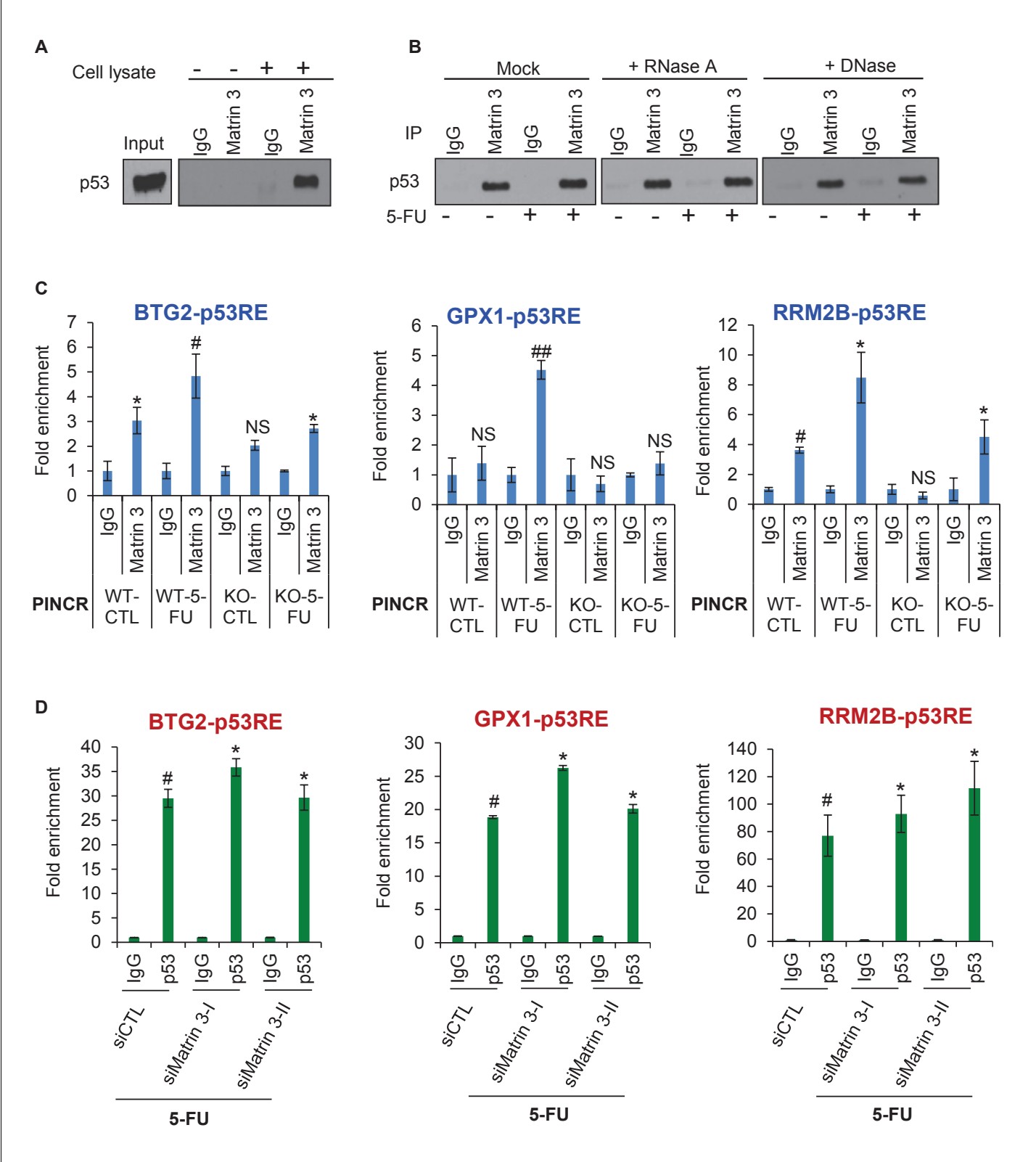

**Figure 6.** Matrin 3 forms a complex with p53 complex and associates with the p53RE of select *PINCR* targets. (**A**) HCT116 cells were treated with 5-FU for 24 hr and immunoblotting for p53 was performed from input, no cell lysate control and IgG or Matrin 3 IP from whole cell extracts. (**B**) HCT116 cells were untreated or treated with 5-FU for 24 hr and the interaction between p53 and Matrin 3 was assessed by IPs from Mock (no extract), RNase-treated or DNase-treated whole cell lysates. (**C**) *PINCR*-WT and *PINCR*-KO cells were untreated or treated with 5-FU for 24 hr and qPCR with primers spanning
*Figure 6 continued on next page*

*Figure 6 continued*

the p53RE of *BTG2*, *RRM2B* and *GPX1* was performed from IgG-ChIP and Matrin 3-ChIP. (**D**) *PINCR*-WT HCT116 cells were reverse transfected with CTL or Matrin 3 siRNAs for 48 hr and then treated with 5-FU for 24 hr. The enrichment of p53 at the p53RE of *PINCR* targets was determined by ChIP-qPCR. Errors bars in C and D represent SD from three independent experiments. *$p<0.05$; #$p<0.01$; ##$p<0.001$.

The following source data and figure supplements are available for figure 6:

**Source data 1.** p53 immunoblot for *Figure 6A and B*.
**Figure supplement 1.** Matrin 3 interacts with p53.
**Figure supplement 1—source data 1.** Matrin 3 immunoblot for *Figure 6—figure supplement 1A*.
**Figure supplement 2.** p53 binding to p21 promoter upon Matrin 3 knockdown.

ChIP-seq data analysis indicates potential chromatin looping between a weak p53 binding region in the enhancer and strong p53 binding region in the promoter of the *PINCR* targets *BTG2*, *GPX1* and *RRM2B*. Notably, whereas the strong ChIP-seq peak at the promoters of these three genes had a canonical p53RE, the weak p53 binding region in their enhancers did not have a canonical p53RE indicating indirect association of p53 at these enhancers.

Next, we sought to determine if Matrin 3 associates with the enhancer of *BTG2*, *GPX1* and *RRM2B* and if this association is dependent on *PINCR*. To test this, we performed ChIP-qPCR for Matrin 3 from *PINCR*-WT and *PINCR*-KO cells after 5-FU treatment. In *PINCR*-WT cells, there was strong enrichment of Matrin 3 at the enhancers of each of these genes (*Figure 7B*). Loss of *PINCR* resulted in significant reduction in Matrin 3 occupancy on each of these enhancer regions (*Figure 7B*). Moreover, after 5-FU treatment, we found significantly reduced p53 binding to these enhancer regions upon loss of *PINCR* (*Figure 7C*) or upon knockdown of Matrin 3 (*Figure 7D*). These results indicate a role of Matrin 3 and *PINCR* in facilitating the association of p53 with the enhancers of specific p53 targets *BTG2*, *GPX1* and *RRM2B* and provide evidence of chromatin looping between the enhancers and promoters of these genes (*Figure 7E*).

### *PINCR* associates with the enhancer regions of select *PINCR* targets via Matrin 3

We next explored the possibility that *PINCR* is also a part of the p53-Matrin 3 complex on the p53RE of *BTG2*, *GPX1* and *RRM2B*. To test this, we used a novel approach in which we tagged endogenous *PINCR* with an S1-tag and utilized the S1-tag to pulldown *PINCR* and then performed qPCR for the p53RE of *BTG2*, *GPX1* and *RRM2B*. The S1-tag is a 44 nt RNA aptamer that binds to streptavidin with high affinity and has been used *in vitro* to identify proteins that bind to S1-tagged RNAs (*Butter et al., 2009*; *Iioka et al., 2011*; *Srisawat and Engelke, 2001, 2002*). To tag endogenous *PINCR*, we used CRISPR/Cas9 and knocked-in a single S1-tag at the 3'end of *PINCR* in HCT116 cells (*Figure 8A* and *Figure 8—figure supplements 1* and *2*). Importantly, the *PINCR-S1* RNA was strongly upregulated (>20 fold) after DOXO treatment (*Figure 8—figure supplement 3A*). Like endogenous untagged *PINCR*, the *PINCR-S1* RNA was predominantly nuclear (*Figure 8—figure supplement 3B*) and expressed at levels comparable to *PINCR* (*Figure 8—figure supplement 3C*). *PINCR* overexpression did not alter the expression of *PINCR-S1* or *BTG2*, *GPX1* and *RRM2B*, suggesting that *PINCR* does not regulate its own expression and that *PINCR* over-expression is not sufficient to alter the expression of *PINCR* targets (*Figure 8—figure supplement 4A and B*).

Streptavidin pulldowns from the *PINCR-S1* expressing cells treated with 5-FU revealed >10-fold enrichment of the *PINCR-S1* RNA (*Figure 8B*) and specific enrichment of Matrin 3 protein (*Figure 8C* and *Figure 8—source data 1*). To determine if *PINCR-S1* associates with the p53RE of *BTG2*, *GPX1* and *RRM2B*, we performed streptavidin pulldowns from formaldehyde-crosslinked parental HCT116 cells (negative control) and the *PINCR-S1* cells after 5-FU treatment. We found that the p53RE of each of these three genes but not the *p21* p53RE, was specifically enriched in the *PINCR-S1* pulldowns (*Figure 8D*). Finally, we examined the association of *PINCR-S1* with the p53RE and enhancer regions of *PINCR* targets in the presence and absence of p53 or Matrin 3. To do this,

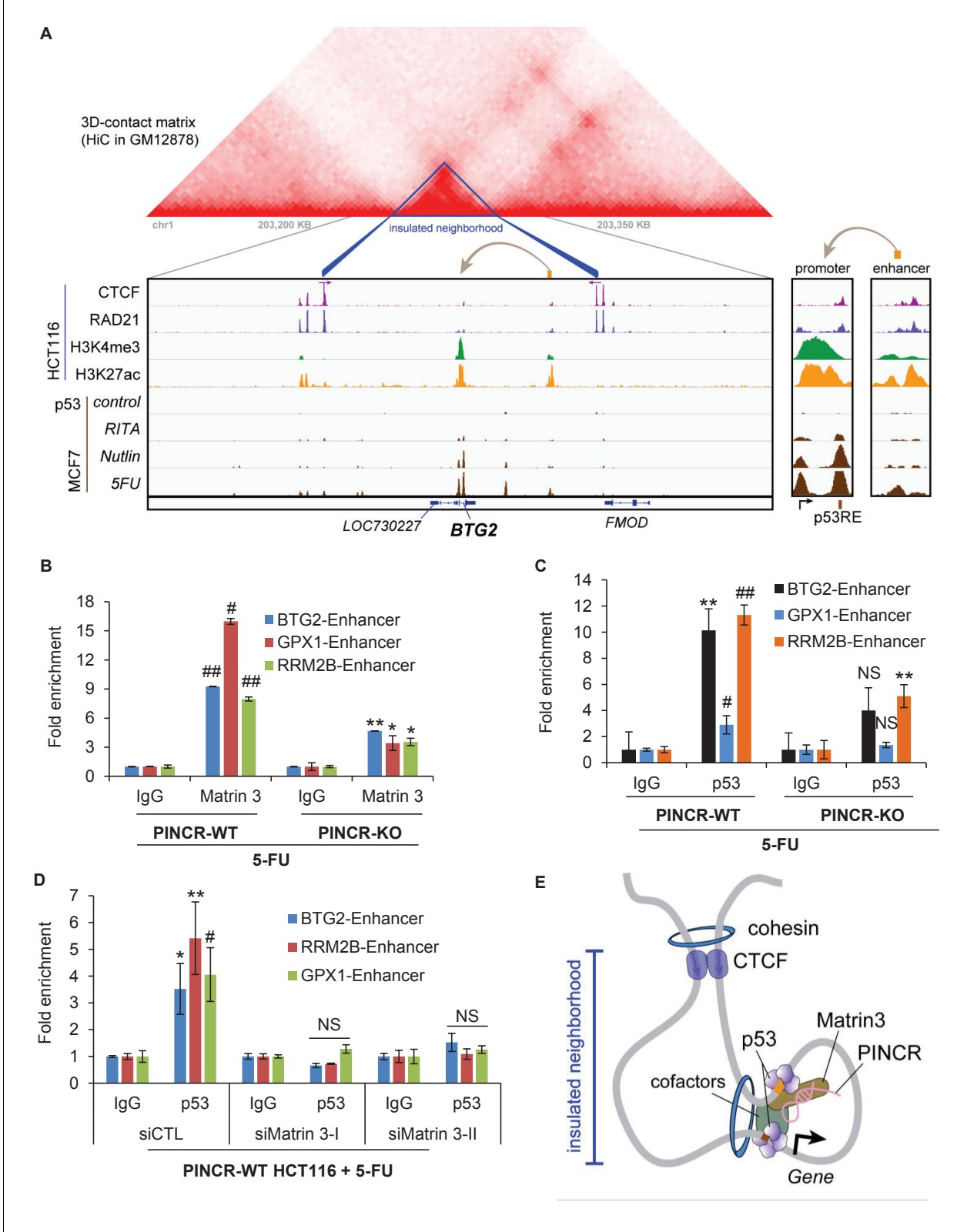

**Figure 7.** *PINCR* modulates the association of Matrin 3 with enhancers of *PINCR* targets within insulated neighborhoods. (**A**) Topological domain and looping structure indicated by 3D contact domain profile (top) surrounding *BTG2* gene and Hi-C data from (*Rao et al., 2014*). ChIP-seq data tracks (middle) in HCT116 showing CTCF anchors and loop-enabling cohesion (RAD21), as well as promoter associated histone mark H3K4me3 and active promoter/enhancer associated mark H3K27ac. p53 ChIP-seq in untreated or treated MCF7 is also shown. Candidate regulatory enhancers (used for

*Figure 7 continued on next page*

*Figure 7 continued*

ChIP-qPCR of Matrin3) are highlighted with an orange box at the tail of an arrow pointing toward the putative target gene. The p53 response element is shown, found at the promoter, where stronger ChIP-seq signal was present upon treatments. The one-dimensional genomic distance between the enhancer and the promoter is indicated between the zoomed in boxes (bottom). (B, C) *PINCR*-WT and *PINCR*-KO HCT116 cells were treated with 5-FU for 24 hr and the association of Matrin 3 (B) or p53 (C) with the enhancer of *PINCR* targets was assessed by ChIP-qPCR. (D) *PINCR*-WT HCT116 cells were reverse transfected with CTL or Matrin 3 siRNAs for 48 hr and then treated with 5-FU for 24 hr. The enrichment of p53 at the enhancer regions of *PINCR* targets was determined by ChIP-qPCR. (E) A cartoon showing the chromatin looping between the enhancer and promoter region of *BTG2*. Errors bars in B, C and D represent SD from three independent experiments. $*p<0.05$; $^{\#}p<0.01$; $**p<0.005$; $^{\#\#}p<0.001$.

The following figure supplements are available for figure 7:

**Figure supplement 1.** Topological domain and looping structure indicated by 3D contact domain profile (top) surrounding *GPX1* gene (chr3:49,200,000–49,500,000) with Hi-C data from (*Rao et al., 2014*).

**Figure supplement 2.** Topological domain and looping structure indicated by 3D contact domain profile (top) surrounding *RRM2B* gene (chr8:103,250,000–103,800,000) with Hi-C data from (*Rao et al., 2014*).

we knocked down p53 (*Figure 8—figure supplement 5* and *Figure 8—figure supplement 5—source data 1*) or Matrin 3 with siRNAs and performed streptavidin pulldowns. As compared to the p53RE in the *PINCR* promoters, in CTL siRNA transfected cells treated with 5-FU, we found stronger association of *PINCR-S1* with the enhancers of the *PINCR* targets (*Figure 8E*). Silencing Matrin 3 or p53 resulted in dramatic reduction of association of *PINCR-S1* with these enhancers and p53REs (*Figure 8E*). Because p53 is important for *PINCR* expression, it is likely that the reduced association of *PINCR-S1* to these regions after p53 knockdown is due to lack of expression. On the other hand, the observed loss in association of *PINCR-S1* to the enhancers and promoters upon knockdown of Matrin 3 indicates that Matrin 3 recruits *PINCR-S1* to these regions. Taken together, these results suggest that a p53-Matrin 3-*PINCR* complex associates with the p53RE and enhancers of *BTG2*, *GPX1* and *RRM2B* and plays a critical role in modulating the induction of these genes after DNA damage.

## Discussion

In this study, we report the first functional characterization of *PINCR*, an intergenic nuclear lncRNA, strongly induced by p53 after DNA damage. Several p53-regulated lncRNAs have been recently identified and shown to play important roles in the p53 network. However, *PINCR* is unique from these recently characterized p53-regulated lncRNAs. Firstly, following p53 activation, the p53-regulated lncRNAs *LED* (*Léveillé et al., 2015*) and *Linc-475* (*Melo et al., 2016*) regulate G1 arrest and prevent entry of cells into mitosis. However, *PINCR*-KO cells show a defect in G1 arrest but the cells arrest in the G2 phase after DNA damage. Secondly, the p53-regulated lncRNAs *lincRNA-p21* (*Dimitrova et al., 2014*), *LED* (*Léveillé et al., 2015*) and *Linc-475* (*Melo et al., 2016*) regulate the levels of *p21*. In addition, in a recent study, the p53-induced lncRNA *DINO*, was shown to directly bind to and regulate p53 levels (*Schmitt et al., 2016*). However, *PINCR* does not alter p53 or p21 levels but instead regulates the expression of the p53 targets *BTG2, GPX1* and *RRM2B* that also regulate G1 arrest after DNA damage.

Our study together with other recent studies shows that specific RNA-binding proteins and transcription factors play an important role in mediating the effects of a lncRNA. For example, in the context of p53 activation, *lincRNA-p21* interacts with hnRNP-K and functions as a coactivator for p53-dependent p21 transcription (*Dimitrova et al., 2014*; *Huarte et al., 2010*). *PANDA*, another p53-regulated lncRNA upstream of *p21*, associates with the transcription factor NF-YA to regulate the expression of pro-apoptotic genes during genotoxic stress (*Hung et al., 2011*). The data presented here indicates that *PINCR* and Matrin 3 act as coactivators of p53 on a subset of p53 targets. It is known that Matrin 3 interacts with enhancer regions (*Romig et al., 1992*; *Skowronska-Krawczyk et al., 2014*), and our data shows that the induction of these genes may be mediated by chromatin looping between Matrin 3 bound to the enhancer regions of these genes and p53 bound to the p53RE in their promoters. *PINCR* recruits Matrin 3 to enhancers of *PINCR*-dependent p53

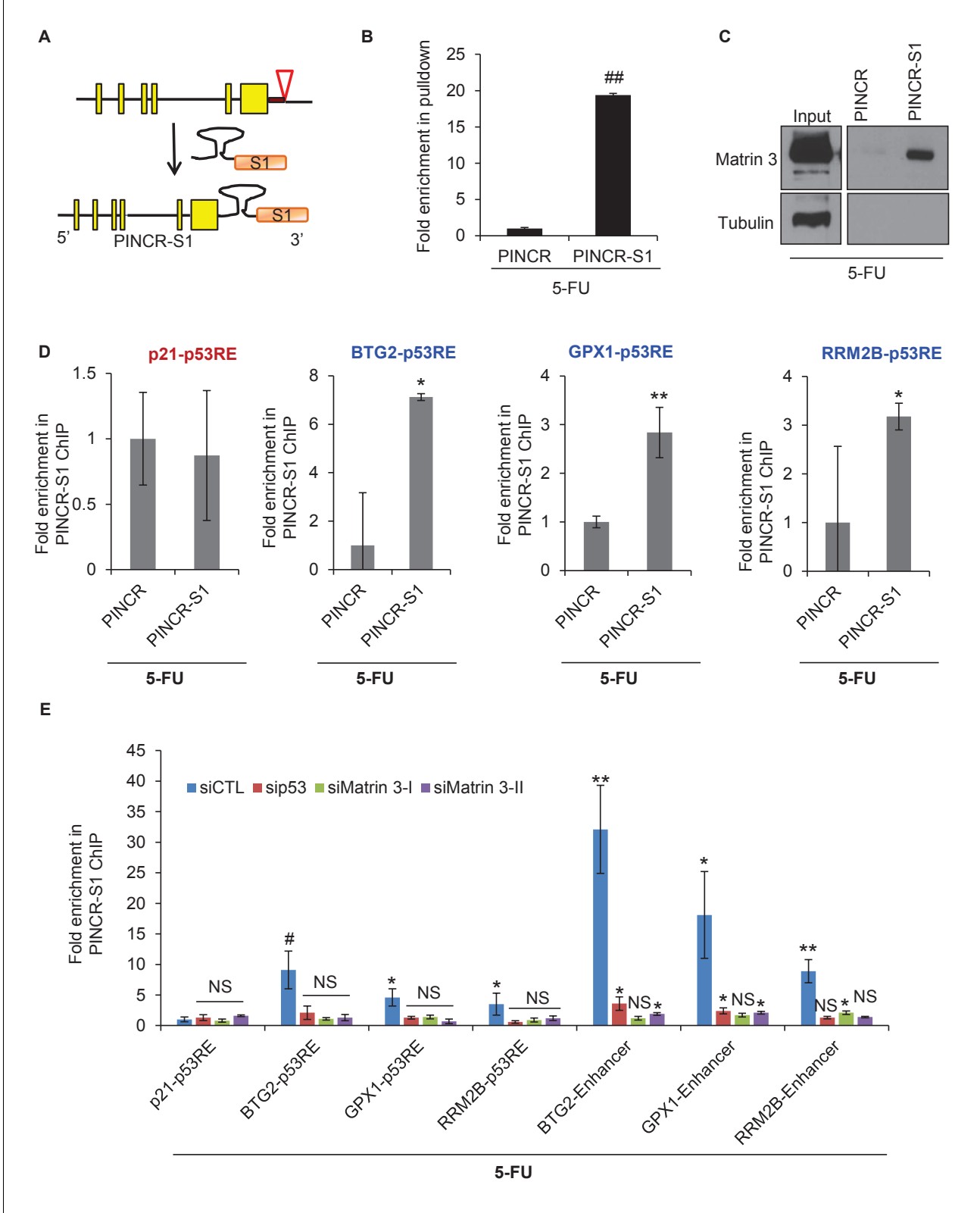

**Figure 8.** *PINCR* associates with the enhancer regions of select p53 targets in a Matrin-3-dependent manner. Schematic showing knock-in of S1-tag at the 3'end of *PINCR*. (**B**) The enrichment of *PINCR* in the streptavidin pulldowns from *PINCR* and *PINCR-S1* cells treated with 5-FU for 24 hr was assessed by qRT-PCR. (**C**) *PINCR* and *PINCR-S1* cells were treated with 5-FU for 24 hr and followed by streptavidin pulldown. Interaction between *PINCR* and Matrin 3 was confirmed by immunoblotting for Matrin 3 or the control Tubulin. (**D**) *PINCR* and *PINCR-S1* cells were treated with 5-FU for 24

*Figure 8 continued on next page*

*Figure 8 continued*

hr and qPCR with primers spanning the p53RE of *BTG2, RRM2B* and *GPX1* was performed following streptavidin pulldown. (E) *PINCR-S1* cells were reverse transfected with CTL siRNA, p53 siRNAs or two independent Matrin 3 siRNAs and then treated with 5-FU for 24 hr. The enrichment of *PINCR-S1* at the p53RE and enhancer regions of *PINCR* targets was determined by ChIP-qPCR from the streptavidin pulldown material. Error bars represent SD from three independent experiments. *p<0.05; **p<0.005, ##p<0.001.

The following source data and figure supplements are available for figure 8:

**Source data 1.** Matrin 3 immunoblot for *Figure 8C*.

**Figure supplement 1.** Sequence alignment of gDNA from *PINCR* (Seq_1) and *PINCR-S1* (Seq_2) clones.

**Figure supplement 2.** Full sequence of the S1 targeting vector.

**Figure supplement 3.** *PINCR-S1* is strongly induced after DNA damage and is a predominantly nuclear lncRNA.

**Figure supplement 4.** Over-expression of *PINCR* in *PINCR-S1* cells does not alter *PINCR-S1* expression or the induction of *PINCR* targets.

**Figure supplement 5.** p53 knockdown in *PINCR*-S1 cells.

**Figure supplement 5—source data 1.** p53 immunoblot for *Figure 8—figure supplement 5*.

target genes. Future studies on the identification of genome-wide-binding sites of Matrin 3 and p53 and epigenetic marks in *PINCR*-WT and *PINCR*-KO cells in the absence or presence of DNA damage will be important. Interestingly, similar intrachromosomal interactions containing enhancer activity have been reported recently and shown to express enhancer RNAs (eRNAs) that are required for efficient transcriptional enhancement of interacting target genes and induction of a p53-dependent cell-cycle arrest (*Léveillé et al., 2015*; *Melo et al., 2013*).

The development of new approaches to identify targets of endogenous lncRNAs is an active area of investigation and remains a major challenge in the lncRNA field. We developed a new approach in which we knocked-in an S1 tag at the 3'end of *PINCR* using CRISPR/Cas9 and determined the association of *PINCR-S1* with the p53RE of specific p53 targets by qPCR following streptavidin pull-downs from crosslinked cells. Our results show a Matrin-3-dependent association of *PINCR-S1* with the enhancer region of *BTG2, RRM2B* and *GPX1* and indicate a direct and specific role of *PINCR* in regulating these genes in response to DNA damage. Given the strong interaction between the S1 tag and streptavidin and studies utilizing transfected S1-tagged RNAs (*Vasudevan and Steitz, 2007*) or in vitro transcribed S1-tagged RNAs (*Butter et al., 2009*; *Iioka et al., 2011*; *Srisawat and Engelke, 2001*, *2002*) for the identification of interacting RNA-binding proteins or miRNAs, this method has the potential to identify the genome-wide targets of *PINCR* and other lncRNAs.

In summary, our study suggests that *PINCR* is an important modulator of gene expression in the p53 pathway that regulates the induction of a subset of p53 targets and this effect is mediated in part via its interaction with Matrin 3. Future investigations on *PINCR* in normal cells and in an expanded panel of cell lines will enhance our understanding of its role in tumorigenesis and tumor progression.

## Materials and methods

### Cell culture, treatments and siRNA transfections

The colorectal cancer cell lines HCT116 (ATCC Number: CCL-247), SW48 (ATCC Number: CCL-231) and RKO (ATCC Number: CRL-2577) and HEK293T (ATCC Number: CRL-11268) cells were purchased from ATCC. The isogenic p53-WT and p53-KO HCT116, RKO and SW48 were previously generated by Bert Vogelstein's lab (Johns Hopkins University). All cell lines were maintained in Dulbecco's modified Eagle's medium (DMEM) (Thermo Fisher Scientific) supplemented with 10% fetal bovine serum (Thermo Fisher Scientific) and 1% penicillin-streptomycin at 37°C, 5% $CO_2$. All cell lines

were routinely checked for mycoplasma using the Venor Gem Mycoplasma detection kit (Sigma-Aldrich, Catalog # MP0025-1KT). Cells were treated with 10 µM Nutlin-3 (Skelleckchem, Catalog # S1061), 300 nM Doxorubicin (DOXO, Catalog # D1515), 100 µM 5-Fluorouracil (5-FU; Calbiochem, Catalog # 343922) or 400 ng/ml NCS (Sigma-Aldrich, Catlog#N9162) for the indicated time.

The Allstars Negative (CTL) siRNAs were purchased from Qiagen and siRNAs for p53 (SMART-pool siRNAs, Catalog # L-003329–00), BTG2 (I-Catalog # J-012308–06 and II-Catalog # J-012308–07), GPX1 (I-Catalog # J-008982–05 and II-Catalog # J-008982–07), RRM2B (I-Catalog # J-010575–05 and II-Catalog # J-010575–06) and Matrin 3 (Catalog # J-017382–05 and J-017382–07) were purchased from Dharmacon. CTL-ASO and *PINCR*-ASO were designed and provided by Ionis Pharmaceuticals (*Supplementary file 5*). All siRNA and ASO transfections were performed by reverse transfection at a final concentration of 20 nM and 50 nM, respectively, using Lipofectamine RNAi-MAX (Life technologies) as directed by the manufacturer. For gene expression analysis after *PINCR* or Matrin 3 knockdown, all the reverse transfections were performed for 48 hr followed by 24 hr DOXO or 5-FU or Nutlin-3 treatment.

## RNA isolation and qRT-PCR

Total RNA from cell lines was isolated using RNeasy mini kit (Qiagen). For qRT-PCR analysis, 500 ng total RNA was reverse-transcribed using iScript Reverse Transcription kit (Bio-Rad), and qPCR was performed using Fast SYBR Green Master Mix (Life technologies) per the manufacturer's instructions. Primer sequences are detailed in *Supplementary file 5*.

## Nuclear and cytoplasmic extract preparations

Nuclear and cytoplasmic extracts were prepared from HCT116 cells expressing *PINCR* or *PINCR-S1*, *PINCR*-WT and *PINCR*-KO, untreated or treated with DOXO or 5-FU for 16 hr or as indicated in figure legend, using Digitonin as previously described (*Lal et al., 2004*). RNA was isolated from cytoplasmic and nuclear fractions using Trizol reagent (Invitrogen) following the manufacturer's protocol.

## Comparative genomic analysis and evaluation of coding potential with PhyloCSF

Strand-specific genomic coordinates for all exons of human *PINCR* and *NEAT1*, *GAPDH*, *SDHA* and *UBC* genes were downloaded from the UCSC Genome Browser (GRCh37/hg19) in BED format. A Multiz alignment of 46 vertebrates aligned to GRCh37/hg19 (http://hgdownload.soe.ucsc.edu/goldenPath/hg19/multiz46way/maf/, in MAF format) was downloaded separately for each gene based on the extracted coordinates for mature transcript accordingly to the UCSC annotation and uploaded to Galaxy (https://usegalaxy.org/). FASTA alignments were generated for each mature transcript separately using «reformat» and «concatenate» options in Galaxy for overlapping list of 29 mammals specified by the PhyloCSF phylogeny (http://mlin.github.io/PhyloCSF/29mammals.nh.png). PhyloCSF was applied to generated FASTA alignments for assessing the coding potential (the Codon Substitution Frequencies score - CSF) of mature transcripts and individual exons of analyzed genes. The CSF score assigns a metric to each codon substitution observed in the input alignment based on relative frequency of that substitution in known coding and non-coding regions. The following parameters were used for analysis: PhyloCSF 29mammals input fasta_file –orf=ATGStop –frames=6 removeRefGaps –aa –allScores. Comparative analysis of all possible reading frames and estimation of the potential to encode any recognizable protein domains was created by BLASTX. Multiple alignments for complete *PINCR* mature transcripts and promoter regions were built using the Muscle program with default parameters (*Edgar, 2004*). Genome rearrangements were analyzed using the Owen program for pair-wise alignments (*Ogurtsov et al., 2002*).

### CRISPR/Cas9-mediated deletion of *PINCR*

gRNAs targeting the 5' and 3' ends of *PINCR* were designed using Zifit software (http://zifit.partners.org/ZiFiT/) and were cloned into U6-gRNA vector (*Moriarity et al., 2014*) having BsmB1 restriction enzyme site. gRNAs sequence information is provided in *Supplementary file 6*. gRNA oligos were ligated and phosphorylated using T4 ligation buffer (NEB) and T4 Polynucleotide Kinase (NEB) using a thermocycler with following parameter: 37°C for 30 min, 95°C for 5 min and then ramp down to 25°C at 5 °C/min. Annealed oligos were ligated with BsmB1 digested U6-gRNA vector (2.9 kb fragment) using quick DNA ligase (NEB). Ligation mix was transformed into *E. coli* DH5-alpha

chemical competent cells and transformants were sequenced to confirm the presence of gRNAs. The efficiency of gRNAs was tested in HEK293T cells by transfecting Cas9 with the gRNAs.

CRISPR-mediated *PINCR*-KO HCT116 or SW48 cells were then generated using *piggyBac* co-transposition method as previously described (*Moriarity et al., 2014*). Cells were cotransfected with 2 µg each of hpT3.5Cagg5-FLAG-hCas9 and the 5′ and 3′ *PINCR* gRNAs cloned in U6-gRNA vector in addition to the 500 ng each of pcDNA-pPB7 transposase and pPBSB-CG-LUC-GFP (Puro)(+CRE) transposon vector using Lipofectamine 2000. After 48 hr, transfected cells were treated with puromycin and incubated at 37°C for 1 week. Cells were then seeded at one cell per well in 96-well plates with puromycin containing DMEM media. Wells that produced single colonies were expanded and DNA was extracted. Clones were then genotyped for deletion of *PINCR* using standard PCR genotyping (*PINCR* deletion analysis primer in *Supplementary file 5*). Identified wild-type (WT) clones were used as controls. PCR products were sequenced to confirm the deletion of *PINCR* genomic locus. Also, total RNA was extracted from individual clones, with and without treatment with DOXO or 5-FU, and expression of *PINCR* was analyzed using qRT-PCR.

## Xenograft assays

Animal protocols were approved by the National Cancer Institute Animal Care and Use Committee following AALAAC guidelines and policies. *PINCR*-WT#1, *PINCR*-KO#1 and *PINCR*-KO#2 cells were untreated or treated with 100 µM 5-FU for 4 hr, following which the drug was washed-off and fresh medium was added. After a 4 hr recovery, live cells were counted with trypan blue exclusion assays and equal numbers of live cells were injected for each sample. Cells ($1 \times 10^6$) were mixed with 30% matrigel in PBS on ice, and the mixture was injected into the flanks of 6- to 8 week-old female athymic nude mice (Animal Production Program, Frederick, MD) (each group $N = 10$). Tumor volume was measured twice a week after 1 week of injection.

To evaluate the effect of proliferation and/or apoptosis in the tumors, the xenograft tumors were collected from four *PINCR*-WT and *PINCR*-KO tumors and fixed in 10% neutral buffered formalin (Sigma, St. Louis, MO). Paraffin sectioning, hematoxylin and eosin staining (H and E), Ki67 staining and Cleaved Caspase-3 staining were performed by Histoserv, Inc (Gaithersburg, MD). The following antibodies were used for immunohistochemistry staining: anti-Ki67 (Abcam, Catalog # Ab16667) and anti-Cleaved Caspase-3 (Cell Signaling, Catalog # 9661). The images were acquired at 40× magnification.

## RNA pulldowns and mass spectrometry

Full-length fragment of *PINCR* was PCR amplified from a pCB6 vector expressing full length *PINCR* using a forward primer containing the T7-promoter sequence at its 5′end and a gene-specific reverse primer (*Supplementary file 5*). The control luciferase cDNA was generated from vector pRL-TK (Addgene) linearized by BamH1 digestion. We then performed *in vitro* transcription to generate biotinylated *PINCR* (Bi-*PINCR*) and the control luciferase (Bi-*LUC*) RNAs using MEGAscript *in vitro* transcription kit (Ambion) and biotin RNA labeling mix (Roche). The *in vitro* transcribed RNA was purified with RNeasy mini kit (Qiagen). The biotinylated RNA was run on the bioanalyzer to check the quality. Nuclear extracts were prepared from HCT116 cells untreated or treated with DOXO for 24 hr as described above. The nuclear lysate was resuspended in RIP buffer (150 mM KCl, 25 mM Tris pH 7.4, 0.5 mM DTT, 0.5% NP40, 1 mM PMSF and protease Inhibitor) and sonicated three times for 5 s and centrifuged at 14,000 x g at 4°C for 30 min. The nuclear lysate was precleared by incubation with Dynabeads M-280 Streptavidin (Thermo Fisher Scientific) for 4 hr at 4°C. In parallel, 40 µl Dynabeads were blocked with 1 mg BSA (company) and 50 µg tRNA (company) for 4 hr at 4°C. Twenty-five pmole Bi-*PINCR* or Bi-*LUC* RNA was incubated with 1 mg precleared nuclear lysate prepared above for 4 hr at 4°C. The biotinylated RNA-protein complexes were pulled down by incubation with preblocked Dynabeads for overnight at 4°C. Interacting proteins were fractionated by SDS-PAGE and each lane cut into 10 slices. The protein bands were then in-gel digested with trypsin (Thermo) overnight at 37°C. The peptides were extracted following cleavage and lyophilized. The dried peptides were solubilized in 2% acetonitrile, 0.5% acetic acid, 97.5% water for mass spectrometry analysis. They were trapped on a trapping column and separated on a 75 µm x 15 cm, 2 µm Acclaim PepMap reverse phase column (Thermo Scientific) using an UltiMate 3000 RSLCnano HPLC (Thermo Scientific). Peptides were separated at a flow rate of 300 nl/min followed by online analysis

by tandem mass spectrometry using a Thermo Orbitrap Fusion mass spectrometer. Peptides were eluted into the mass spectrometer using a linear gradient from 96% mobile phase A (0.1% formic acid in water) to 55% mobile phase B (0.1% formic acid in acetonitrile) over 30 min. Parent full-scan mass spectra were collected in the Orbitrap mass analyzer set to acquire data at 120,000 FWHM resolution; ions were then isolated in the quadrupole mass filter, fragmented within the HCD cell (HCD normalized energy 32%, stepped ±3%), and the product ions analyzed in the ion trap. Proteome Discoverer 2.0 (Thermo) was used to search the data against human proteins from the UniProt database using SequestHT. The search was limited to tryptic peptides, with maximally two missed cleavages allowed. Cysteine carbamidomethylation was set as a fixed modification, and methionine oxidation set as a variable modification. The precursor mass tolerance was 10 ppm, and the fragment mass tolerance was 0.6 Da. The Percolator node was used to score and rank peptide matches using a 1% false discovery rate.

## Generation of S1-tagged *PINCR* line

For *PINCR* tagging, we used integration by Non-homologous end joining, which was accomplished by introducing a simultaneous double-strand break in genomic DNA and in the targeting vector at the 5' of the S1-tag (*Brown et al., 2016*). Plasmids encoding spCas9 and sgRNAs were obtained from Addgene (Plasmids #41815 and #47108). Oligonucleotides for construction of sgRNAs were obtained from Integrated DNA Technologies, hybridized, phosphorylated and cloned into the sgRNA plasmid or targeting vector using BbsI sites (*Brown et al., 2017*). Target sequences for sgRNAs are provided in *Supplementary file 6*.

We prepared the targeting vector by first synthesizing two complementary oligonucleotides (IDT) with the sequence of the S1 tag followed by the sequence of *RP3-326I13.1* located at the 3' of the sgRNA-binding site, which potentially contains the native elements for termination of transcription. The oligonucleotides were dimerized, phosphorylated and cloned into the targeting vector using T4 ligase. Subsequently, we introduced at the 5' of the S1 tag the sequence targeted by the *RP3-326I13.1* sgRNA. The targeting vector also contained an independent PuroR expression cassette driven by a PGK promoter for facile isolation of clonal populations of cells that integrate the plasmid within the genome. The sequence of the targeting vector is provided in *Figure 8—figure supplement 2*. HCT116 cells were transfected with 300 ng Cas9, 300 ng of sgRNA and 300 ng of targeting vector using Lipofectamine 2000 (Invitrogen) according to the manufacturer's instructions in 24-well plates. Three days after transfection, the cells were selected with 0.5 µg/ml Puromycin to generate clonal populations.

Genomic DNA from each clone was isolated using DNEasy Blood and Tissue Kit (Qiagen). PCRs to detect integration of the targeting vector at the target site were performed using KAPA2G Robust PCR kits (Kapa Biosystems) according to the manufacturer's instructions. A typical reaction contained 20–100 ng of genomic DNA in Buffer A (5 µl), Enhancer (5 µl), dNTPs (0.5 µl), primers forward (*PINCR* Det FP, 1.25 µl) and reverse (Targeting vector Det RP, 1.25 µl) and KAPA2G Robust DNA Polymerase (0.5 U). The DNA sequences of the primers for each target are provided in *Supplementary file 5*. PCR products were visualized in 2% agarose gels and images were captured using a ChemiDoc-It2 (UVP). The PCR products were cloned into TOPO-TA cloning (ThermoFisher) and sequenced.

## RNA pulldowns from S1-tagged *PINCR* cells

For immunoprecipitation experiments, HCT116 clonal cell lines expressing *PINCR* or *PINCR-S1* RNA were treated with 5-FU for 24 hr to induce *PINCR* expression. $2 \times 10^7$ cells were lysed in lysis buffer (150 mM NaCl, 50 mM Tris-HCl pH 7.5, 0.5% Triton X-100, 1 mM PMSF, Protease inhibitor cocktail and RNase inhibitor). Lysates were sonicated three times for 5 s and centrifuged at 14,000 x g at 4°C for 30 min. For IP, 500 µg of cellular extract was incubated overnight at 4°C with 25 µl Dynabeads M-280 Streptavidin (Thermo Fisher Scientific). Beads were washed twice with high salt buffer (0.1% SDS, 1% Triton-X-100, 2 mM EDTA, 20 mM Tris-HCl pH 8 and 500 mM NaCl) followed by low salt buffer (0.1% SDS, 1% Triton-X-100, 2 mM EDTA, 20 mM Tris-HCl pH 8 and 150 mM NaCl) and TE buffer (10 mM Tris-HCl pH 8 and 2 mM EDTA). Bound proteins were eluted by boiling the samples for 5 min in SDS-PAGE sample buffer. Eluted proteins were subjected to SDS-PAGE and immunoblotting with Matrin 3 (Bethyl labs) or β-Tubulin (Cell Signaling, Catalog # 2146S). Enrichment of

*PINCR* RNA levels in the pulldown material was evaluated by directly adding Trizol to the beads, followed by RNA extraction and qRT-PCR.

To test the binding of *PINCR* to the chromatin, *PINCR* and *PINCR-S1* cells were treated with 5-FU for 24 hr to induce *PINCR* expression. Chromatin was cross-linked with 1% formaldehyde, and cells were lysed and sonicated in Buffer B (1% SDS, 10 mM EDTA, 50 mM Tris-HCl pH 8, Protease inhibitor and RNase inhibitor). RNA-DNA-protein complexes were immunoprecipitated with Dynabeads M-280 Streptavidin, overnight using IP buffer (0.01% SDS, 0.5% Triton-X-100, 1.2 mM EDTA, 16.7 mM Tris-HCl pH 8 and 167 mM NaCl). Beads were washed twice with high-salt buffer followed by TE buffer. Bound RNA-DNA-protein complexes were eluted from the beads using elution buffer (100 mM NaCl, 50 mM Hepes pH 7.4, 0.5% NP40, 10 mM $MgCl_2$ and 5 mM Biotin), at room temperature for 20 min. Eluted material was incubated at 65°C for 2 hr (200 mM NaCl) to reverse crosslink the bound proteins. The samples were treated with Proteinase K and eluted DNA was column purified (Qiagen) and analyzed by qPCR using primers flanking the p53-binding sites of different genes (*Supplementary file 5*).

To test if *PINCR* binding to the chromatin is p53 and/or Matrin-3-dependent, *PINCR-S1* cells were reverse transfected with CTL siRNAs, p53 siRNAs or two independent Matrin 3 siRNAs. After 48 hr, the cells were treated with 5-FU for 24 hr. The enrichment of *PINCR-S1* at the promoter and enhancer regions of *PINCR* targets was determined by ChIP-qPCR followed by streptavidin pulldown as described above.

## Microarray analysis

For lncRNA profiling HCT116, SW48 and RKO cells were untreated or treated with Nutlin-3 in duplicate for 8 hr. Total RNA was isolated using the RNeasy Mini kit (Qiagen) and hybridized to Affymetrix HT2.0 arrays that contain probes for ~11,000 lncRNAs. To identify the *PINCR*-regulated transcriptome *PINCR*-WT and *PINCR*-KO cells were untreated or treated with 5-FU (100 uM) for 24 hr. RNA samples were prepared as described above in triplicates and labeled using the IlluminaToTalPrep RNA amplification kit (Ambion) and microarrays were performed with the HumanHT-12 v4 Expression BeadChip kit (Illumina). After hybridization, raw data were extracted with Illumina GenomeStudio software. Raw probe intensities were converted to expression values using the lumi package in Bioconductor with background correction, variance stabilization and quantile normalization. Differential expression between different conditions was computed by an empirical Bayes analysis of a linear model using the limma package in Bioconductor. Adjusted p-values were calculated with the Benjamini and Hochberg method, and differentially expressed genes were selected with adjusted p-value≤0.05 and a fold change ≥1.50.

All the microarray data for this study has been deposited in GEO. The Accession number is GSE90086 and the URL is https://www.ncbi.nlm.nih.gov/geo/query/acc.cgi?acc=GSE90086. The unpublished RNA-seq data (Li et al., unpublished) used in this study has been deposited in GEO. The Accession number is GSE79249 and the URL is https://www.ncbi.nlm.nih.gov/geo/query/acc.cgi?acc=GSE79249.

## Luciferase reporter assays

A 2 kb region upstream of the first exon of *PINCR* was PCR amplified (primer sequences in *Supplementary file 5*) using 100 ng genomic DNA from HCT116 cells and inserted into upstream of *Firefly* luciferase of pGL3 luciferase vector (Promega). To measure *PINCR* promoter activity, HCT116 cells were co-transfected with 100 ng of pGL3-empty vector or pGL3 expressing the *PINCR* promoter, along with pCB6-empty vector or pCB6 expressing p53, and 10 ng pRL-TK expressing *Renilla* luciferase. After 48 hr, luciferase activity was measured using the dual-luciferase reporter system (Promega).

A 2 kb *PINCR* promoter region (chrX: 43,034,255–43,036,255) with (WT-p53RE) and without (Δp53RE) the 20 bp p53RE (GCCCTTGTCTGGACATGCCC) was synthesized in pGL3 luciferase vector by GenScript. HCT116 cells were co-transfected with 100 ng of pGL3 expressing the WT or Δp53RE *PINCR* promoter and 10 ng pRL-TK expressing *Renilla* luciferase. After 48 hr, cells were left untreated or treated with DOXO for 24 hr and luciferase activity was measured using the dual-luciferase reporter system (Promega).

## Flow cytometry, caspase-3 immunostaining, colony formation and cell proliferation assays

For cell cycle analysis, $3.0 \times 10^5$ *PINC*R-WT and *PINC*R-KO cells were seeded per well of a 6-well plate. After 24 hr, cells were untreated or treated with 300 nM DOXO or 100 µM 5-FU or 400 ng/ml NCS or 10 µM Nutlin-3 and the samples were collected at the indicated times. Cells were fixed with ice-cold ethanol for 2 hr and stained with propidium iodide (Sigma) in the presence of RNase A (Qiagen). Cell cycle profiles were captured using FACS Calibur flow cytometer (BD Biosciences), and the data were analyzed using FlowJo software (FloJo, LLC).

To perform cell cycle analysis after Matrin 3 knockdown, *PINC*R-WT and *PINC*R-KO cells were reverse transfected with siCTL and two independent Matrin 3 siRNAs using RNAiMAX at a final siRNA concentration of 20 nM. After 48 hr, cells were untreated or treated with 300 nM DOXO or 100 µM 5-FU and cell cycle profiles were captured as described above. For cell cycle analysis after BTG2/GPX1/RRM2B knockdown, *PINC*R-WT HCT116 cells were reverse transfected with siCTL and two individual siBTG2, siGPX1 and siRRM2B using RNAiMAX at a final siRNA concentration of 20 nM. After 48 hr, cells were untreated or treated with 100 µM 5-FU and cell cycle profiles were captured as described above. Cell cycle analysis after *PINCR* knockdown using ASOs, HCT116 cells were reverse transfected with 50 nM CTL-ASO or PINCR-ASO. After 48 hr, cells were untreated or treated with 300 nM DOXO or 100 µM 5-FU and cell cycle profiles were captured as described above.

For caspase-3 immunostaining, $3.0 \times 10^5$ *PINC*R-WT and *PINC*R-KO cells were seeded per well of a six-well plate. After 24 hr, cells were untreated or treated with DOXO for 72 hr and fixed with 4% paraformaldehyde for 10 min and permeabilized by 0.5% Triton X-100 for 10 min. Fixed cells were stained for 1 hr with primary antibodies anti-Mab414 (Covance, Catalog # MMS120P) for nuclear envelope and active caspase-3 (Cell Signaling, Catalog # 9661S) for apoptotic cells, followed by further staining with DAPI (blue) and secondary antibodies, anti-mouse 586 (orange; Alexa Fluor 586 goat anti-mouse IgG, Life Technology, Catalog # A11031) and anti-rabbit 488 (green; Alexa Fluor 488 donkey anti-rabbit IgG, life Technology, Catalog # A21206) for 1 hr. Images were taken by Ziess immunofluorescence microscope with x63 lens.

For colony formation on plastic, $3 \times 10^5$ *PINC*R-WT and *PINC*R-KO HCT116 cells were seeded per well in six-well plates. After 24 hr, cells were untreated or treated with 100 nM or 300 nM DOXO or 10 µM, 50 µM or 100 µM 5-FU for 4 hr, following which the drug was washed-off and fresh medium was added. After a 4 hr recovery, cells were seeded in a 12-well plate at a density of 500 cells per well. After 2 weeks, colonies were fixed with ice-cold 100% methanol for 5 min, stained with crystal violet, and colonies were counted. For colony formation after ASO transfections, HCT116 cells were transfected with CTL-ASO and PINCR-ASO. After 48 hr, cells were untreated or treated with 100 µM 5-FU for 4 hr, following which the drug was washed-off and fresh medium was added. After a 4 hr recovery, cells were seeded in a 12-well plate at a density of 500 cells per well. After 2 weeks, colonies were fixed as described above. For long-term cell proliferation assays on plastic, $3 \times 10^5$ *PINC*R-WT and *PINC*R-KO SW48 cells were seeded in 12-well plate. After 24 hr, cells were untreated or treated with 100 µM 5-FU for 7 days. After 7 days, cells were fixed with ice-cold 100% methanol for 5 min and stained with crystal violet.

## *PINCR* overexpression lines and cell cycle analysis

The 2.2 kb transcript corresponding to *PINCR* RNA (NR_110387.1) was cloned into pCB6 vector using EcoR1/Xba1 restriction enzyme cloning sites. *PINCR*-KO cells were transfected with pCB6 empty vector (EV) or pCB6 vector expressing *PINCR*. After 48 hr, the cells were treated with neomycin and incubated at 37°C for 4–5 days, for selection of stably transfected cells. Total RNA was extracted from the pool of cells and expression of *PINCR* was analyzed using qRT-PCR. For cell cycle analysis, $3.0 \times 10^5$ *PINC*R-KO cells expressing pCB6-EV or pCB6-*PINCR* were seeded per well in six-well plates. After 24 hr, cells were untreated or treated with 300 nM DOXO and the samples were collected at indicated times. FACS analysis was performed as described above.

To determine the effect of *PINCR* overexpression on cell cycle and gene regulation, HCT116 cells expressing *PINCR* or *PINCR-S1* were transfected with pCB6 or pCB6-*PINCR* expressing *PINCR*. After 48 hr, cells were treated with DOXO or 5-FU for indicated times. Expression of *PINCR* and other target genes was analyzed using qRT-PCR and FACS was performed as described above.

## Immunoblotting and immunoprecipitation

To measure apoptosis after 5-FU treatment *PINCR*-WT and *PINCR*-KO cells were untreated or treated with 5-FU (100 µM) for 48 hr. Similarly, to determine the levels of p53 and/or p21 or phospho-Rb in *PINCR*-WT and *PINCR*-KO cells, the cells were untreated or treated with 5-FU for 24 hr. Whole-cell lysates were prepared using radioimmunoprecipitation (RIPA) buffer containing protease inhibitor cocktail (Roche). Proteins were quantified using the bicinchoninic acid protein quantitation (BCA) kit (Thermo Scientific). For immunoblotting, 20 µg whole cell lysate per lane was loaded onto a 12% SDS-PAGE gel, transferred to nitrocellulose membrane and immunoblotted with anti-Cleaved PARP (Cell Signaling, Catalog # 2541), anti-p53 (DO-1) (Santa Cruz, Catalog # sc-126), anti-p21 (Santa Cruz, Catalog # sc-397), anti-Histone H3 (Cell Signaling, Catalog # 4620), anti-phospho-Rb (Cell Signaling, Catalog # 9307P, 9208P, 9301P) and anti-GAPDH (Cell Signaling, Catalog # 14C10).

To determine the p53 and Matrin 3 knockdown efficiency *PINCR*-WT cells were reverse transfected with CTL siRNAs or p53 siRNAs or Matrin 3 siRNAs (20 nM) respectively, and 48 hr after transfection cell lysates were prepared using RIPA buffer as described above, followed by immunoblotting with anti-p53, Matrin 3 (Bethyl labs Catalog # A300-591A) or GAPDH antibodies.

For co-immunoprecipitation experiments HCT116 cells were untreated or treated with 5-FU (100 µM) for 24 hr and whole cell lysates were prepared in RIPA buffer and centrifuged at 14,000 x g at 4°C for 30 min. For IP, 25 µl Pierce protein A/G magnetic beads (Thermo Scientific, Catalog # 88802) were incubated with 2 µg Matrin 3 antibody or IgG control (Santa Cruz, Catalog # sc-2027), for 4 hr at 4°C. Following this, 500 µg cellular extract was incubated for 4 hr at 4°C with A/G magnetic beads pre-coated with IgG or Matrin 3. Beads were washed 5 times at 4°C with RIPA buffer and samples were untreated or treated with RNase A and DNase for 30 min at 37°C. Bound proteins were eluted by boiling the samples for 5 min in SDS-PAGE sample buffer. Eluted proteins were subjected to SDS-PAGE and immunoblotting using anti-p53 DO-1 antibody. To perform reciprocal IP nuclear lysates were prepared from HCT116 cells as described before and Immunoprecipitation was done as discussed above. IgG control (Santa Cruz, Catalog # sc-2025) and DO-1 p53 antibodies were used for IP and anti-Matrin 3 antibody for immunoblotting.

To determine the association of *PINCR* to Matrin 3 in intact cells, $2 \times 10^7$ HCT116 cells were treated with 5-FU (100 µM) for 24 hr and then cross-linked with 1% formaldehyde. Crosslinked cells were resuspended in Buffer B (1% SDS, 10 mM EDTA, 50 mM Tris-HCl pH 8, Protease inhibitor cocktail and RNAse inhibitor), followed by sonication. An aliquot of the sonicated cell lysates was subjected to IP using 2 µg IgG or Matrin 3 antibodies for 4 hr at 4°C on protein A/G magnetic beads, using IP buffer (0.01% SDS, 1.1% Triton X-100, 1.2 mM EDTA, 16.7 mM Tris-HCl pH 8, 167 mM NaCl). The IP material was washed twice with high salt buffer (0.1% SDS, 1% Triton-X-100, 2 mM EDTA, 20 mM Tris-HCl pH 8 and 500 mM NaCl) followed by TE buffer (10 mM Tris-HCl pH 8 and 2 mM EDTA). Bound RNA-protein complexes were eluted from the beads using elution buffer (0.1% SDS, 0.1M NaHCO$_3$, RNase inhibitor), at 37°C for 15 min followed by reverse cross-linking at 65°C for 2 hr by with 200 mM NaCl. Matrin 3 bound RNAs were isolated by phenol-chloroform extraction (Ambion) followed by ethanol precipitation and qRT-PCR was used to determine the enrichment of *p21* (negative control) and *PINCR* in the Matrin 3 IPs.

To determine the direct binding of *PINCR* to recombinant Matrin 3 (rMatrin 3), 200 ng of in vitro transcribed Bi-*PINCR* or Bi-*LUC* RNA was incubated with 500 ng recombinant Matrin 3 protein (Creative BioMart, Catalog # MATR3-15H) in 1X EMSA buffer (25 mM Tris-HCl pH 7.5, 150 mM KCl, 0.1% Triton-X-100, 100 µg/ml BSA, 2 mM DTT and 5% glycerol) at room temperature for 2 hr. RNA–protein complex was immunoprecipitated at room temperature for 2 hr, by using Dynabeads M-280 Streptavidin. Beads were washed five to six times with 1X EMSA buffer without glycerol and bound material was subjected to SDS-PAGE and immunoblotting for Matrin3. The following antibodies were used: anti-p53 (DO-1) 1: 1000 dilution from Santa Cruz Biotechnology; anti-Matrin 3 1:1000 dilution from Bethyl Laboratories; anti-Cleaved PARP and anti-GAPDH at 1:1000 dilution from Cell Signaling.

## Chromatin IP assays

Chromatin IP was performed with the Active Motif ChIP kit (Active Motif, Carlsbad, CA, USA) as directed by the manufacturer. Briefly, $5 \times 10^7$ HCT116 cells grown in 15-cm plates were untreated or treated with DOXO (300 nM) for 16 hr. Similarly, *PINCR*-WT and *PINCR*-KO cells were untreated

or treated with 5-FU (100 µM) for 24 hr. Chromatin was cross-linked with 1% formaldehyde, and cells were lysed and sonicated. Protein–DNA complexes were immunoprecipitated with control IgG or anti-p53 (DO1) (Santa Cruz) or anti-Matrin 3 (Bethyl labs) antibody. The IP material was washed and heated at 65°C overnight to reverse crosslinks. ChIP DNA was column purified (Qiagen) and analyzed by qPCR. Primers flanking the p53 binding sites or the enhancer regions of different genes are listed in *Supplementary file 5*. To test if association of p53 to the promoter and enhancer regions is Matrin-3-dependent $5 \times 10^7$ *PINCR*-WT cells were reverse transfected with CTL siRNAs and two independent Matrin 3 siRNAs. After 48 hr, cells were treated with 5-FU for 24 hr and enrichment of p53 at the promoter and enhancer regions of *PINCR* targets was determined by ChIP-qPCR as described above.

## Bioinformatic analysis of ChIP-seq and Hi-C data

Integrative Genome Browser (IGV, software.broadinstitute.org/software/igv/) was used to download and visualize relevant ChIP-seq datasets from the ENCODE consortium data repository. Motif identification at CTCF peaks surrounding putative enhancer-gene pairs within insulated neighborhoods was gathered from the HOMER software package (http://homer.ucsd.edu/homer/ngs/). Genomic locations of p53 response elements were determined similarly from HOMER p53 motif datasets. Chromatin folding was inferred from 3D contact matrices calculated from in situ HiC data (*Rao et al., 2014*) and visualized using the Juicebox desktop application (*Durand et al., 2016*).

## Acknowledgements

We thank Tom Misteli, Glenn Merlino, Susan Gottesman, Shiv Grewal, Curtis Harris and Javed Khan for their critical comments on the manuscript and Bert Vogelstein for the isogenic cell lines. This research was supported by the Intramural Research Program (AL, LW, LMJ) of the National Cancer Institute (NCI), Center for Cancer Research (CCR), NIH. KVP lab is supported by grants from NIH [GM088252] and American Cancer Society [RSG-11-174-01-RMC].

## Additional information

### Funding

| Funder | Grant reference number | Author |
|---|---|---|
| National Cancer Institute | NIH IRP | Ashish Lal |

The funders had no role in study design, data collection and interpretation, or the decision to submit the work for publication.

### Author contributions

RC, AL, Conceptualization, Writing—original draft, Writing—review and editing; BG, WSW, MM, MD, BSM, Methodology; MS, XLL, Validation; MFJ, YY, Investigation; LMJ, Validation, Methodology; SAS, LMW, YZ, Formal analysis; SMF, Resources; KVP, Resources, Validation; PP-P, Methodology, Writing—review and editing

### Author ORCIDs

Ashish Lal, http://orcid.org/0000-0002-4299-8177

### Ethics

Animal experimentation: This study was performed in strict accordance with the recommendations in the Guide for the Care and Use of Laboratory Animals of the National Institutes of Health. All animal studies were conducted under protocol LC-070 approved by the Animal Care and Use Committee of the National Cancer Institute, The Frederick National Laboratory and the Center for Cancer Research are accredited by AALAC International and follow the Public Health Service Policy for the Care and Use of Laboratory Animals.

# Additional files

## Supplementary files

• Supplementary file 1. Transcripts induced upon Nutlin treatment of parental HCT116, RKO and SW48 cells.

• Supplementary file 2. LncRNAs induced after Nutlin treatment in HCT116, RKO and SW48 cells.

• Supplementary file 3. Microarray analysis from PINCR-WT and PINCR-KO cells untreated or treated with 5-FU (100 uM). 'KO' refers to PINCR-KO and WT refers to PINCR-WT. 'FC' refers to fold change. The 11 genes regulated by PINCR and also p53 are shown in red.

• Supplementary file 4. Mass spectrometry analysis from RNA pulldowns. PSM refers to peptide-spectrum match. Proteins enriched at least twofold in the PINCR pulldowns as compared to Luciferase pulldowns under untreated condition (Unt) and Doxorubicin treatment (DOXO) are shown. To obtain a non-zero fold change a value of '1' was assigned to the PSM if it was zero.

• Supplementary file 5. Sequence of primers.

• Supplementary file 6. Sequence of guide RNAs.

## Major datasets

The following datasets were generated:

| Author(s) | Year | Dataset title | Dataset URL | Database, license, and accessibility information |
|---|---|---|---|---|
| Yuelin Jack Zhu | 2017 | Prosurvival long noncoding RNA PINCR regulates a subset of p53 targets in colorectal cancer cells by binding to Matrin 3 | https://www.ncbi.nlm.nih.gov/geo/query/acc.cgi?acc=GSE90086 | Publicly available at NCBI Gene Expression Omnibus (accession no: GSE90086) |
| Lal A, Wen X | 2017 | Identification of long noncoding RNAs regulated by p53 | https://www.ncbi.nlm.nih.gov/geo/query/acc.cgi?acc=GSE79249 | Publicly available at NCBI Gene Expression Omnibus (accession no: GSE79249) |

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
