## [Decision Letter]

[Editors’ note: the authors were asked to provide a plan for revisions before the editors issued a final decision. What follows is the editors’ letter requesting such plan.]

Thank you for sending your article entitled "Prosurvival long noncoding RNA PINCR regulates a subset of p53 targets in colorectal cancer cells by binding to Matrin 3" for peer review at *eLife*. Your work is being evaluated by a Senior Editor and three reviewers, one of whom is a member of our Board of Reviewing Editors.

We are in the process of discussing the reviews. Before advising further and reaching a decision, we would like to hear your response to the following concerns, along with an estimated time frame for completing any additional work (please see the reviews below).

*Reviewer #1:*

This is a tantalizing paper reporting the existence of p53-inducible lncRNA, dubbed PINCR, that is supposed to act as a gene-specific cofactor for induction of select p53 target genes, such as BTG2, GPX1 and RRM2B, but not the potent CDK inhibitor p21. In the absence of PINCR, the authors observe and impairment in G1 arrest upon p53 activation with genotoxic agents, concurrently with increased apoptosis. Mechanistically, PINCR is required for binding of p53 to the enhancers of some of the genes it regulates (e.g. BTG2, RRM2B, GPX1). However, PINCR does not bind to p53 directly, but binds instead to the RNA-binding protein Matrin-3. In turn, Matrin-3 is required for p53 transactivation of the genes that require PINCR. In sum, the authors conclude that the PINCR-Matrin complex is a gene-specific cofactor of p53, required for p53 binding to select enhancers.

Overall, the paper is tantalizing and interesting, but there are many unresolved aspects that prevent its publication in its current form. There are several areas where more clarity is needed about the observations made, as some of them contradict well known facts in the p53 field.

The major concerns are:

1) A p21-independent role of PINCR in p53-dependent cell cycle arrest. The authors show that PINCR is required for p53-dependent G1 arrest in response to doxorubicin, neocarcinostatin and 5FU. The authors also show that PINCR does not affect p21 expression or binding of p53 to the p21 enhancer. This is highly paradoxical, because p21 is required for G1 arrest in response to the three p53-activating agents mentioned above. The results counter this well-established and reproducible observation, and suggest that p21 cannot enforce G1 arrest without transactivation of other p53 targets, such as BTG2, RRM2B and GPX1. If this is the case, it should be demonstrated thoroughly. Can depletion of BTG2 and/or RRM2B and/or GPX1 recapitulate the effects of PINCR depletion? Is it true that these genes are required for p21-dependent cell cycle arrest? This could be answered with siRNA knockdowns. Is it true that p21 is unable to block CDKs, prevent RB phosphorylation and stall the cell cycle in the absence of PINCR? This should be explored in detail by looking a p21 localization, Rb phosphorylation and other markers of p21 action and G1-S progression.

2) A gene-specific role for PINCR and Matrin. A key observation in this paper is that PINCR and Matrin are required for p53 transactivation at selective loci, and this is explained by a requirement of PINCR for p53 binding to the respective enhancers at those genes. The notion of a RNA (or a RNA binding protein) being required for p53 binding to some DNA elements but not others is very provocative and requires further investigation. Biochemically speaking, p53 is a potent DNA-binding protein that works as a tetramer to bind a 4-repeat DNA sequence (a 20 mer consensus made of two half sites with two palindromic repeats per half site). How is this activity modified by RNA and/or a RNA-binding protein at some DNA elements but not others? This is very tantalizing and hard to comprehend. In vitro protein-DNA binding experiments should be performed to investigate this. The genome-wide requirement for PINCR and Matrin should be defined. Is it true that only 11 p53 targets require PINCR? If so, is it true that PINCR regulates p53 binding at only those 11 loci? Is it true that PINCR binds only to those p53 target loci? What about Matrin? What is Matrin's binding pattern relative to p53 target genes, those sensitive and insensitive to PINCR?

These questions should be answered with careful ChIP-seq experiments for p53, PINCR-S1 and Matrin, and a detailed investigation of this 'gene-specific effects'. This is important, because the latest and most comprehensive investigations of p53 functionality at enhancers indicate that p53 does not employ cofactors to recognize and activate transcription from ~1000 enhancers in a wide variety of contexts (see for example Verfaillie et al., Genome Research 2016). This notion is further supported by earlier work demonstrating that p53 can recognize its p53REs in nucleosomal context, often leading to nucleosome displacement.

3) DNA- versus RNA-dependent effects of PINCR. The authors conclude that PINCR acts by a RNA-dependent mechanism. However, this is based mostly on an unsatisfactory rescue overexpression experiment in PINCR knockout cells. This is important, because the field has been profoundly misled by earlier reports of another p53-inducible lncRNA, lincRNA-p21, which was first described to be acting by an RNA-dependent mechanism, yet now is ample clear that the lincRNA-p21 locus acts by a DNA-dependent, enhancer-like mechanism (see latest work from John Rinn's team in Cell Reports, backtracking on his original Cell paper by Huarte et al). To clarify this for PINCR, the authors should elaborate more on their knockout. What exact region of the genome did they delete when they deleted the 'entire PINCR region'? Did they delete the proximal p53 binding site as well? If they preserved the p53 enhancer, they should demonstrate that p53 binding to that region is intact by ChIP.

Other concerns:

1) Repeats and error bars in 2B.

2) Is PINCR upregulated in published GRO-seq data in HCT116 cells? It is unclear which ncRNA in Allen et al. the authors refer too. Is it refereed by a different name in Allen et al? A genome viewer screenshot will go a long way to show that PINCR is truly a direct p53 target is truly direct.

3) Raw cell cycle profiles for 3B-C.

4) To confirm that PINCR is involved in p53-dependent G1 arrest, they should repeat the experiments they did with genotoxic agents using Nutlin instead.

*Reviewer #2:*

This manuscript by Chaudhary et al. describes the characterization and functional analysis of a new p53-induced lncRNA, designated PINCR. The authors demonstrated that PINCR is a direct p53 target gene that is necessary for the induction of a subset of p53 targets involved in cell cycle regulation and apoptosis. Evidence is presented supporting a model in which a PINCR-Matrin 3 complex facilitates p53 association with select promoters. The identification of a lncRNA that is essential for transactivation of specific p53 targets and, as a result, whose loss impairs the p53 pathway is a significant finding and, in principle, appropriate for publication in *eLife*. However, there are several aspects of the work that are incomplete and require further experimentation to convincingly support the conclusions put forward.

1) Since this is the first functional analysis of PINCR, additional basic characterization of the lncRNA is important. The authors should confirm the 5'/3' ends and splicing of the transcript by RACE and RT-PCR. The copy number of the expressed transcript should be determined (e.g. +/- Nutlin, DOXO, and/or 5-FU). A northern blot would be helpful if possible to confirm the existence of a discrete transcript at the expected size. The conservation pattern of the transcript and p53 response element should be commented upon.

2) The use of genome editing to stably delete PINCR is a good approach to establish a robust system for functional studies. However, the authors chose to delete a ~50 kb segment to remove the lncRNA which raises concerns regarding whether other important sequences were removed in addition to the 2.2 kb spliced PINCR sequence. I appreciate the authors' attempts to mitigate these concerns by performing a rescue experiment but the analysis of rescued cells does not go nearly far enough. Cell cycle analysis of rescued cells should be shown in addition to the provided analysis of cell survival after DOXO treatment (Figure 3). The clonogenic survival assays (Figure 3) should also be performed on rescued cells. Likewise, cell cycle, cell survival, and clonogenic survival of rescued cells after 5-FU treatment should be added to Figure 4.

Any concerns about the genome editing approach could be fully mitigated if the authors employed an orthogonal method to inhibit PINCR such as siRNA knockdown or CRISPRi, followed by analysis of cell cycle, survival, clonogenic growth, and target gene induction (e.g. BTG2, GPX1, RRM2B).

3) While p53-mediated PINCR induction was demonstrated in multiple cell lines, all functional analyses are limited to HCT116. PINCR should be knocked out or inhibited in a second cell line to confirm that the phenotypic effects observed are not unique to HCT116.

4) Knockdown of Matrin 3 partially phenocopies PINCR knockout with respect to cell survival after 5-FU treatment (Figure 6). As this is a very important prediction of their model, the analysis of Matrin 3 knockdown should be extended. First, at least 2 independent siRNAs should be tested to avoid off-target effects (this is especially important when monitoring general cellular phenotypes such as survival). Furthermore, the cell cycle should be examined after knockdown of Matrin 3 and 5-FU treatment and both cell cycle and apoptosis should be examined after DOXO treatment.

5) What happens to the Matrin-p53 interaction in PINCR KO cells? Co-IP assays (as in Figure 7) should be performed in these cells. Likewise, does PINCR associate with p53 response elements in the absence of Matrin and/or p53? The experiments in Figure 7 should be repeated after Matrin and p53 knockdown.

6) While the observation that p53-mediated induction of BTG2, GPX1, and RRM2B is attenuated in PINCR KO cells and after Matrin 3 knockdown is interesting, there is no evidence that these changes in gene expression are responsible for the impaired p53-mediated cell cycle arrest and apoptosis in PINCR KO cells. Does knockdown of any of these targets phenocopy PINCR KO? Does overexpression of any of these genes rescue PINCR KO?

7) The p53-RE luciferase assays in Figure 2 would be much more convincing if a reporter with mutations in the p53-RE were tested in parallel as a negative control.

8) In Figure 7—figure supplement 1, it appears that the polyadenylation signal was removed when the S1 tag was inserted (at the end of the red sequence). Does this affect the level of PINCR expression? Although the authors show PINCR-S1 induction after DOXO treatment, PINCR-S1 levels should be directly compared to endogenous PINCR.

*Reviewer #3:*

In this original study, the authors identify the lincRNA PINCR as a new direct transcriptional target of p53. They further show that PINCR is required for p53-mediated G1 arrest, owing to the contribution of PINCR to the induction of a specific subset of p53 target genes in response to DNA damage. Importantly, they identify an interaction between PINCR and the RNA binding protein matrin 3, which is required for the induction of this subset of genes, and show nicely that PINCR binds directly to the p53 response elements (p53 RE's) of those genes.

Overall, this is a very interesting and well performed study. Publication in *eLife* is recommended if the authors can address, in a satisfactory manner, several important issues as listed below.

1).Figure 4. In view of the suggested biology of PINCR, one might expect it to be selectively LESS induced under conditions that favor p53-dependent cell death. What happens, for example, upon treatment of HCT116 cells with high dose 5-FU? As a matter of fact, the data in Figure 4 and its supplemental already point in this direction: 5-FU, at the concentration used by the authors, induces more apoptosis (11%) than the other treatments, while causing a relatively modest increase in PINCR. Thus, the relative extent of endogenous PINCR induction might be a determinant of the likelihood of the cells to die rather than arrest. I suggest that the authors address this possibility more closely, e.g. by employing several different concentrations of 5-FU and comparing the extent of PINCR induction, induction of a typical pro-apoptotic p53 target gene (e.g. PUMA or CD95), and the accrual of sub-G1 cells.

2) Figure 4. The in vivo result is impressive. However, it does not necessarily support a pro-survival function of PINCR as concluded in the subsection “PINCR loss results in hypersensitivity to 5-FU and decreased tumor growth”: the authors did not address survival/death at all in this experiment. It will be important to perform histopathological examination of the tumors, and show (by cleaved caspase 3 or TUNEL) that the PINCR KO tumors indeed manifest a greater extent of apoptosis.

3) I could not find any ChIP experiment showing that matrin 3 is required for optimal binding of p53 to the p53RE's of BTG2 etc. Did I miss it? This is an important experiment, as the authors attribute the reduced expression of those genes to reduced recruitment of p53 to their p53RE's.

4) Why does PINCR bind only to a subset of p53RE's? Is there significant homology between the sequence of PINCR and that of the DNA in the vicinity of those p53RE's, but not the p21 p53RE for example?

5) Figure 3: How does PINCR overexpression affect the cell cycle profile and the expression of the specific subset of G1-related genes without and with DNA damage? Is PINCR overexpression alone sufficient to upregulate those genes?

6) Figure 7: What happens to matrin 3 after DNA damage? Do its levels increase? Does it become more localized to the nucleus?

7) Does PINCR regulate its own expression after DNA damage? Does PINCR knockout affect the binding of p53 to the PINCR promoter?

8) Figure 7. Does matrin 3 bind those promoters also without DNA damage, or is it recruited only after DNA damage?

9) Does matrin 3 affect the induction of BTG2 etc. in response to Nutlin, or is this DNA damage-specific?

[Editors’ note: formal revisions were requested, following approval of the authors’ plan of action.]

Thank you for submitting your article "Prosurvival long noncoding RNA PINCR regulates a subset of p53 targets in colorectal cancer cells by binding to Matrin 3" for consideration by *eLife* and for sending us your plan for revisions. The plan was favourably assessed by the Reviewing Editor and the reviewers and we are prepared to request a submission of a revised manuscript along the lines of what is outlined in your plan. The evaluation of your submission has been overseen by a Reviewing Editor and Kevin Struhl as the Senior Editor. The following individual involved in review of your submission has agreed to reveal his identity: Moshe Oren (Reviewer #3).

In your revisions, please also address the following concerns:

1) Response #1 to Reviewer 2: Since the RNA-seq data matches the annotated transcript ends, 5'/3' RACE is not necessary. But the data should be shown in the revised manuscript.

2) Response #6 to Reviewer 2: Use of the ASO approach is encouraged to bolster the genome editing findings. ASOs would actually be preferable to CRISPRi in this case because if ASO transfection phenocopies genomic deletion of PINCR, this would provide strong evidence for an RNA-mediated function of the transcript. CRISPRi would not rule out the possibility that the observed phenotypes are somehow the result of transcription of this locus, rather than a function of the RNA itself.

3) Response #8 to Reviewer 2: Two independent siRNAs are needed. A pool of siRNAs can still result in off-target effects.

4) ChIP-seq experiments for Matrin 3 and PINCR-S1 are no longer necessary.

5) The p53 ChIP-seq experiments are still requested, even if this extends the resubmission timeline between 2-3 months.

---

## [Author Response]

[Editors’ note: what follows is the authors’ plan to address the revisions.]

We thank the reviewers for their constructive comments and suggestions to improve our paper. All 3 reviewers found our paper interesting. We believe that we can address all of the major concerns of reviewers 2 and 3. Reviewer 1 has made some excellent suggestions and we can carefully address most of his/her concerns. However, to be realistic and to carefully conduct the ChIP-seq experiments (suggested by Reviewer 1) on Matrin 3 and PINCR-S1, it may take several months; thus, these experiments are beyond the scope of this study. As an alternative approach, we have proposed a series of experiments that will prove our model according to which *PINCR* regulates a subset of p53 targets by binding to Matrin 3.

Reviewer #1:

This is a tantalizing paper reporting the existence of p53-inducible lncRNA, dubbed PINCR, that is supposed to act as a gene-specific cofactor for induction of select p53 target genes, such as BTG2, GPX1 and RRM2B, but not the potent CDK inhibitor p21. In the absence of PINCR, the authors observe and impairment in G1 arrest upon p53 activation with genotoxic agents, concurrently with increased apoptosis. Mechanistically, PINCR is required for binding of p53 to the enhancers of some of the genes it regulates (e.g. BTG2, RRM2B, GPX1).

We would like to clarify that in our paper, we showed that PINCR is required for binding of p53 to the p53RE in the promoters of these genes (Line 256). Regulation of binding of p53 to the enhancers of these genes via Matrin 3 and PINCR may be one mechanism by which PINCR acts, a point that we had mentioned in the Discussion but if we are given an opportunity to revise this paper, we will test this as mentioned (please see: response # 5 to reviewer #1).

However, PINCR does not bind to p53 directly, but binds instead to the RNA-binding protein Matrin-3. In turn, Matrin-3 is required for p53 transactivation of the genes that require PINCR. In sum, the authors conclude that the PINCR-Matrin complex is a gene-specific cofactor of p53, required for p53 binding to select enhancers.

Overall, the paper is tantalizing and interesting, but there are many unresolved aspects that prevent its publication in its current form. There are several areas where more clarity is needed about the observations made, as some of them contradict well know facts in the p53 field.

We thank this reviewer for appreciating our findings and the constructive comments. Below is our response and plans to address the unresolved issues.

The major concerns are:

1) A p21-independent role of PINCR in p53-dependent cell cycle arrest. The authors show that PINCR is required for p53-dependent G1 arrest in response to doxorubicin, neocarcinostatin and 5FU. The authors also show that PINCR does not affect p21 expression or binding of p53 to the p21 enhancer. This is highly paradoxical, because p21 is required for G1 arrest in response to the three p53-activating agents mentioned above. The results counter this well-established and reproducible observation, and suggest that p21 can not enforce G1 arrest without transactivation of other p53 targets, such as BTG2, RRM2B and GPX1. If this is the case, it should be demonstrated thoroughly. Can depletion of BTG2 and/or RRM2B and/or GPX1 recapitulate the effects of PINCR depletion? Is it true that these genes are required for p21-dependent cell cycle arrest? This could be answered with siRNA knockdowns. Is it true that p21 is unable to block CDKs, prevent RB phosphorylation and stall the cell cycle in the absence of PINCR? This should be explored in detail by looking a p21 localization, Rb phosphorylation and other markers of p21 action and G1-S progression.

This is an excellent point. As the reviewer pointed out, p21 plays an important role in mediating the effects of p53 when it comes to G1 arrest after DNA damage. According to some papers, in the absence of p21, there is no G1 arrest after DNA damage (Waldman et al., Cancer Res, 1995). However, according to other papers (also cited in the Waldman et al. paper), p21 is not the only player (Deng et al., Cell, 1995). In addition, there are other papers in support of both models. Our data shows that the% of cells arrested in G1 after 24 hr (Figure 3) is 28-35% in PINCR-WT clones and 10-15% in PINCR-KO clones. However, we found no difference in p21 mRNA induction upon loss of PINCR.

This was an unexpected result and indicates, as pointed out by the reviewer, that p21 cannot enforce G1 arrest if BTG2, RRM2B and GPX1 are not induced. Although this may seem paradoxical, it may be a truly novel finding – p21 may need these 3 genes to enforce G1 arrest. As suggested by the reviewer, we will test the roles of these genes in p21-dependent G1 arrest by siRNA knockdown of p21, and these 3 genes. This experiment will give us a sense of the contribution of each of these genes to the G1 arrest and their significance to p21 biology. In addition, as suggested, we will look at p21 localization because as mentioned by the reviewer, it may be the case that loss of PINCR alters p21 localization leading to altered Rb phosphorylation. In addition, we will look at other markers of p21 action and G1/S progression, as suggested by this reviewer.

These are straightforward experiments that can be addressed in a timely manner and the results will be instrumental in determining the role of these 3 PINCR targets in G1 arrest after DNA damage and their role in in p53 and p21 biology.

2) A gene-specific role for PINCR and Matrin. A key observation in this paper is that PINCR and Matrin are required for p53 transactivation at selective loci, and this is explained by a requirement of PINCR for p53 binding to the respective enhancers at those genes. The notion of a RNA (or a RNA binding protein) being required for p53 binding to some DNA elements but not others is very provocative and requires further investigation. Biochemically speaking, p53 is a potent DNA-binding protein that works as a tetramer to bind a 4-repeat DNA sequence (a 20 mer consensus made of two half sites with two palindromic repeats per half site). How is this activity modified by RNA and/or a RNA-binding protein at some DNA elements but not others? This is very tantalizing and hard to comprehend. In vitro protein-DNA binding experiments should be performed to investigate this. The genome-wide requirement for PINCR and Matrin should be defined. Is it true that only 11 p53 targets require PINCR?

Our data indicates that PINCR regulates only 11 p53 targets. This result is consistent with our additional data (to be added to the revised paper) that PINCR is expressed at 19-38 copies per HCT116 cell after DNA damage. A lncRNA expressed at 19-38 copies per cell is likely not to regulate all p53 targets, until it regulates p53 itself, which is not the case.

If so, is it true that PINCR regulates p53 binding at only those 11 loci? Is it true that PINCR binds only to those p53 target loci? What about Matrin? What is Matrin's binding pattern relative to p53 target genes, those sensitive and insensitive to PINCR?

Our data (Figure 7, Figure 7—figure supplement 1 and Figure 7)) shows that Matrin 3 and PINCR associate specifically with the p53RE of BTG2, RRM2B and GPX1 but not the p53RE of p21. We can extend this analysis to the remaining 8 out of 11 p53 targets. The results will further establish the role of PINCR and Matrin 3 in regulation of this subset of p53 targets. In addition, the proposed PINCR-S1 pulldown experiments after knockdown of p53 or Matrin 3 (please see: response # 10 to reviewer # 2) will support these conclusions.

These question should be answered with careful ChIP-seq experiments for p53, PINCR-S1 and Matrin, and a detailed investigation of this 'gene-specific effects'. This is important, because the latest and most comprehensive investigations of p53 functionality at enhancers indicate that p53 does not employ cofactors to recognize and activate transcription from ~1000 enhancers in a wide variety of contexts (see for example Verfaillie et al., Genome Research 2016). This notion is further supported by earlier work demonstrating that p53 can recognize its p53REs in nucleosomal context, often leading to nucleosome displacement.

The reviewer’s suggestion of conducting ChIP-seq experiments for p53, PINCR- S1 and Matrin 3 will allow detailed gene-specific effects of PINCR and Matrin 3. We can do the p53 ChIP-seq from PINCR-WT and PINCR-KO cells ± 5-FU. The results will further support our model of gene-specific regulation by PINCR.

The suggestion to do ChIP-seq for Matrin 3 and PINCR-S1 is great. However, unlike p53 ChIP-seq, the conditions for Matrin 3 ChIP-seq will need to be optimized. Moreover, to make sure that the ChIP-seq peaks are bona fide Matrin 3 associated sites, the experiments will also have to be done from Matrin 3 knockdown and/or Matrin 3 knockout cells. Likewise, the PINCR-S1 ChIP-seq will require optimization and scaling up because even for the very abundant lncRNAs such as MALAT1, NEAT1 and HOTAIR, endogenous lncRNA pulldown methods such as ChIRP, RAP and CHART require 50-100 million cells. Thus, it will take several months to get high confidence Matrin 3 and PINCR-S1 ChIP-seq data, if done carefully. These experiments are therefore beyond the scope of this study but we plan to follow-up on this.

*How is this activity modified by RNA and/or a RNA-binding protein at some DNA elements but not others?*

This is an intriguing question. A likely scenario is that Matrin 3 binds to enhancer regions of these genes and regulates p53 occupancy of their promoter (see Discussion,). To address this question, we hypothesize that Matrin 3 and PINCR associate with the enhancer regions upstream of BTG2, RRM2B and GPX1. We have looked at potential p53 binding sites in the enhancer regions for these genes (and negative controls such as p21) and will examine the association of p53, Matrin 3, PINCR-S1 and epigenetic marks in these regions in PINCR-WT and PINCR-KO cells by ChIP-qPCR.

3) DNA- versus RNA-dependent effects of PINCR. The authors conclude that PINCR acts by a RNA-dependent mechanism. However, this is based mostly on an unsatisfactory rescue overexpression experiment in PINCR knockout cells. This is important, because the field has been profoundly misled by earlier reports of another p53-inducible lncRNA, lincRNA-p21, which was first described to be acting by an RNA-dependent mechanism, yet now is ample clear that the lincRNA-p21 locus acts by a DNA-dependent, enhancer-like mechanism (see latest work from John Rinn's team in Cell Reports, backtracking on his original Cell paper by Huarte et al). To clarify this for PINCR, the authors should elaborate more on their knockout. What exact region of the genome did they delete when they deleted the 'entire PINCR region'? Did they delete the proximal p53 binding site as well? If they preserved the p53 enhancer, they should demonstrate that p53 binding to that region is intact by ChIP.

We used 2 gRNAs that were designed upstream of the first and downstream of the last exon on PINCR, respectively. Based on Sanger sequencing data, we found that out of the 2 PINCR-KO clones used in our study, one clone has intact p53RE whereas the other clone has partially intact p53RE. We can provide the DNA sequence showing the exact region of the genome that was deleted. As suggested, we can check the binding of p53 to the p53RE of PINCR in both PINCR-KO clones. We would like to mention that unlike lincRNA-p21 that is encoded upstream of p21, PINCR is expressed from an intergenic region from the X-chromosome and none of the 11 p53 PINCR-regulated genes are on the X-chromosome. Therefore, unlike lincRNA-p21, PINCR is a trans-acting lncRNA.

Other concerns:

1) Repeats and error bars in 2B.

We will repeat this experiment to generate error bars.

*2) Is PINCR upregulated in published GRO-seq data in HCT116 cells? It is unclear which ncRNA in Allen et al. the authors refer too. Is it refereed by a different name in Allen et al? A genome viewer screenshot will go a long way to show that PINCR is truly a direct p53 targetis truly direct.*

The reference Allen et al. was cited to determine how many of the protein-coding genes in our lncRNA array data (Figure 1) were also identified in the p53 GRO-seq paper. Genome viewer screenshot of p53 ChIP-seq data (Figure 2), the sequence of the p53RE (Figure 2—figure supplement 1) and the data in Figure 2 indicate that PINCR is a direct target of p53. In addition, as suggested by reviewer # 3 (please see: response # 3 to reviewer # 3), we will mutate the p53RE in the PINCR promoter that was inserted upstream of luciferase (Figure 2) and perform luciferase assays.

*3) Raw cell cycle profiles for 3B-C.*

We will add these profiles to supplemental material.

*4) To confirm that PINCR is involved in p53-dependent G1 arrest, they should repeat the experiments they did with genotoxic agents using Nutlin instead.*

As suggested, we will do this experiment to further support the role of PINCR in p53-dependent G1 arrest.

Reviewer #2:

This manuscript by Chaudhary et al. describes the characterization and functional analysis of a new p53- induced lncRNA, designated PINCR. The authors demonstrated that PINCR is a direct p53 target gene that is necessary for the induction of a subset of p53 targets involved in cell cycle regulation and apoptosis. Evidence is presented supporting a model in which a PINCR-Matrin 3 complex facilitates p53 association with select promoters. The identification of a lncRNA that is essential for transactivation of specific p53 targets and, as a result, whose loss impairs the p53 pathway is a significant finding and, in principle, appropriate for publication in eLife. However, there are several aspects of the work that are incomplete and require further experimentation to convincingly support the conclusions put forward.

We thank this reviewer for appreciating our findings and the constructive comments. Below is our response and plans to address his/her comments.

1) Since this is the first functional analysis of PINCR, additional basic characterization of the lncRNA is important. The authors should confirm the 5'/3' ends and splicing of the transcript by RACE and RT-PCR.

This is a very important point. We have done RNA-seq data from HCT116, RKO and SW48 p53+/+ and p53-/- cells (Li et al., unpublished). This data can be used to show the 5’ and 3’ends of this transcript. The RNA-seq data matches the annotated 5’ and 3’ends of PINCR. This can be further supported by RT-PCR. If the reviewer wants, we can also do 5’ and 3’ RACE.

*The copy number of the expressed transcript should be determined (e.g. +/- Nutlin, DOXO, and/or 5-FU).*

We have found that after DNA damage with DOXO, PINCR is expressed at 19- 38 copies per cell. This data will be added to the revised paper.

*A northern blot would be helpful if possible to confirm the existence of a discrete transcript at the expected size.*

We tried the Northern blot but it did not work. This is not surprising given our finding that PINCR is expressed at 19-38 copies per cell.

The conservation pattern of the transcript and p53 response element should be commented upon.

We will add this information in the revised paper.

2) The use of genome editing to stably delete PINCR is a good approach to establish a robust system for functional studies. However, the authors chose to delete a ~50 kb segment to remove the lncRNA which raises concerns regarding whether other important sequences were removed in addition to the 2.2 kb spliced PINCR sequence. I appreciate the authors' attempts to mitigate these concerns by performing a rescue experiment but the analysis of rescued cells does not go nearly far enough. Cell cycle analysis of rescued cells should be shown in addition to the provided analysis of cell survival after DOXO treatment (Figure 3). The clonogenic survival assays (Figure 3) should also be performed on rescued cells. Likewise, cell cycle, cell survival, and clonogenic survival of rescued cells after 5-FU treatment should be added to Figure 4.

We have the cell cycle data from the rescue experiments and will add it to the revised paper. Alternatively, these issues can be addressed by PINCR knockdown with antisense oligos (ASOs) and/or CRISPRi as suggested below.

Any concerns about the genome editing approach could be fully mitigated if the authors employed an orthogonal method to inhibit PINCR such as siRNA knockdown or CRISPRi, followed by analysis of cell cycle, survival, clonogenic growth, and target gene induction (e.g. BTG2, GPX1, RRM2B).

Excellent point. When we initiated this study, we tried to knockdown PINCR with siRNAs but we did not get robust knockdown. However, we now have antisense oligos (ASOs) and CRISPR-KRAB expressing HCT116 cells that can be used to knockdown PINCR and examine the phenotype and target gene induction, as suggested by the reviewer. We are confident that this approach will successfully knockdown PINCR and the results will further solidify our findings from the PINCR-KO clones.

3) While p53-mediated PINCR induction was demonstrated in multiple cell lines, all functional analyses are limited to HCT116. PINCR should be knocked out or inhibited in a second cell line to confirm that the phenotypic effects observed are not unique to HCT116.

This is an important point. We will knockdown PINCR in another cell line with ASOs or CRISPRi and determine if the observed phenotypic effects are not restricted to HCT116.

4) Knockdown of Matrin 3 partially phenocopies PINCR knockout with respect to cell survival after 5-FU treatment (Figure 6). As this is a very important prediction of their model, the analysis of Matrin 3 knockdown should be extended. First, at least 2 independent siRNAs should be tested to avoid off-target effects (this is especially important when monitoring general cellular phenotypes such as survival). Furthermore, the cell cycle should be examined after knockdown of Matrin 3 and 5-FU treatment and both cell cycle and apoptosis should be examined after DOXO treatment.

In our paper, we have used smart pool siRNAs that contain 4 siRNAs. If the reviewer insists, we will do the experiments with 2 independent siRNAs. In addition, we will perform the cell cycle and apoptosis after Matrin 3 knockdown as suggested by the reviewer.

5) What happens to the Matrin-p53 interaction in PINCR KO cells? Co-IP assays (as in Figure 7) should be performed in these cells.

Our data (Figure 7) indicates that p53-Matrin 3 interaction is RNA-independent. So, we expect p53 to interact with Matrin 3 even in PINCR-KO cells. However, it would be good to do this co- IP experiment in PINCR-KO cells, as suggested by the reviewer.

Likewise, does PINCR associate with p53 response elements in the absence of Matrin and/or p53? The experiments in Figure 7 should be repeated after Matrin and p53 knockdown.

Great suggestion. The suggested experiments can be done easily and will further establish the model and also answer reviewer #1 and # 3 concern.

6) While the observation that p53-mediated induction of BTG2, GPX1, and RRM2B is attenuated in PINCR KO cells and after Matrin 3 knockdown is interesting, there is no evidence that these changes in gene expression are responsible for the impaired p53-mediated cell cycle arrest and apoptosis in PINCR KO cells. Does knockdown of any of these targets phenocopy PINCR KO?

A similar point was raised by reviewer # 1 (please see: response # 1 to reviewer # 1). We will address this by knocking down BTG2, GPX1 and RRM2B alone or in combination and determine the effect on cell cycle arrest and apoptosis, as suggested.

Does overexpression of any of these genes rescue PINCR KO?

We can attempt this experiment but if overexpression of these genes in the absence of DNA damage causes cell cycle arrest, it may be complicate the analysis.

7) The p53-RE luciferase assays in Figure 2 would be much more convincing if a reporter with mutations in the p53-RE were tested in parallel as a negative control.

We agree with the reviewer and will be do this experiment.

8) In Figure 7—figure supplement 1, it appears that the polyadenylation signal was removed when the S1 tag was inserted (at the end of the red sequence). Does this affect the level of PINCR expression? Although the authors show PINCR-S1 induction after DOXO treatment, PINCR-S1 levels should be directly compared to endogenous PINCR.

We can address this question by RT-qPCR from cells expressing PINCR or PINCR-S1.

Reviewer #3:

In this original study, the authors identify the lincRNA PINCR as a new direct transcriptional target of p53. They further show that PINCR is required for p53-mediated G1 arrest, owing to the contribution of PINCR to the induction of a specific subset of p53 target genes in response to DNA damage. Importantly, they identify an interaction between PINCR and the RNA binding protein matrin 3, which is required for the induction of this subset of genes, and show nicely that PINCR binds directly to the p53 response elements (p53 RE's) of those genes.

Overall, this is a very interesting and well performed study. Publication in eLife is recommended if the authors can address, in a satisfactory manner, several important issues as listed below.

We thank this reviewer for his/her kind words and constructive comments. Below is our response and plans to address the reviewer’s comments.

1) Figure 4. In view of the suggested biology of PINCR, one might expect it to be selectively LESS induced under conditions that favor p53-dependent cell death. What happens, for example, upon treatment of HCT116 cells with high dose 5-FU? As a matter of fact, the data in Figure 4 and its supplemental already point in this direction: 5-FU, at the concentration used by the authors, induces more apoptosis (11%) than the other treatments, while causing a relatively modest increase in PINCR. Thus, the relative extent of endogenous PINCR induction might be a determinant of the likelihood of the cells to die rather than arrest. I suggest that the authors address this possibility more closely, e.g. by employing several different concentrations of 5-FU and comparing the extent of PINCR induction, induction of a typical pro-apoptotic p53 target gene (e.g. PUMA or CD95), and the accrual of sub-G1 cells.

This is a very interesting point. As suggested, at several different concentrations of 5-FU, we will measure the induction of PINCR, PUMA and CD95 and also determine the effect on G1 arrest and the sub-G1 population.

*2) Figure 4. The* in vivo *result is impressive. However, it does not necessarily support a pro-survival function of PINCR as concluded in the subsection “PINCR loss results in hypersensitivity to 5-FU and decreased tumor growth”: the authors did not address survival/death at all in this experiment. It will be important to perform histopathological examination of the tumors, and show (by cleaved caspase 3 or TUNEL) that the PINCR KO tumors indeed manifest a greater extent of apoptosis.*

This is an important point. We have the frozen tissues and will compare apoptosis between the injected PINCR-WT and PINCR-KO cells as suggested.

3) I could not find any ChIP experiment showing that matrin 3 is required for optimal binding of p53 to the p53RE's of BTG2 etc. Did I miss it? This is an important experiment, as the authors attribute the reduced expression of those genes to reduced recruitment of p53 to their p53RE's.

We agree with the reviewer. This experiment was also suggested by reviewer # 2 (please see: response #10 to reviewer # 2) and will test the binding of p53 to the p53RE of the PINCR targets after Matrin 3 knockdown.

4) Why does PINCR bind only to a subset of p53RE's? Is there significant homology between the sequence of PINCR and that of the DNA in the vicinity of those p53RE's, but not the p21 p53RE for example?

This point was also raised by reviewer # 1. Please see our response to reviewer # 1 (response #2-4 to reviewer # 1).

5) Figure 3: How does PINCR overexpression affect the cell cycle profile and the expression of the specific subset of G1-related genes without and with DNA damage? Is PINCR overexpression alone sufficient to upregulate those genes?

Another great suggestion. This is a straightforward experiment that will test if PINCR overexpression is sufficient to upregulate these genes and to determine the effect on G1 arrest after DNA damage.

6) Figure 7: What happens to matrin 3 after DNA damage? Do its levels increase? Does it become more localized to the nucleus?

As suggested, we will measure Matrin 3 protein levels and nuclear localization after DNA damage. Preliminary data suggests that Matrin 3 expression is not deregulated after DNA damage.

7) Does PINCR regulate its own expression after DNA damage? Does PINCR knockout affect the binding of p53 to the PINCR promoter?

We will check the effect of overexpressing exogenous PINCR on PINCR-S1 cells using primers specific to the S1-tagged lncRNA. As mentioned before, the PINCR-KO cells have a complete or partial loss of the p53RE in the PINCR promoter. We will test the binding of p53 to the p53RE, as mentioned in the response to reviewer 1 (response # 6 to reviewer # 1).

8) Figure 7. Does matrin 3 bind those promoters also without DNA damage, or is it recruited only after DNA damage?

We will determine by ChIP-qPCR this in the revised paper.

9) Does matrin 3 affect the induction of BTG2 etc in response to Nutlin, or is this DNA damage-specific?

We will address this in the revised paper by knocking down Matrin 3 and treating the cells with Nutlin.

[Editors’ notes: the authors’ response after being formally invited to submit a revised submission follows.]

*Reviewer #1:*

*Overall, the paper is tantalizing and interesting, but there are many unresolved aspects that prevent its publication in its current form. There are several areas where more clarity is needed about the observations made, as some of them contradict well known facts in the p53 field.*

*The major concerns are:*

*1) A p21-independent role of PINCR in p53-dependent cell cycle arrest. The authors show that PINCR is required for p53-dependent G1 arrest in response to doxorubicin, neocarcinostatin and 5FU. The authors also show that PINCR does not affect p21 expression or binding of p53 to the p21 enhancer. This is highly paradoxical, because p21 is required for G1 arrest in response to the three p53-activating agents mentioned above. The results counter this well-established and reproducible observation, and suggest that p21 cannot enforce G1 arrest without transactivation of other p53 targets, such as BTG2, RRM2B and GPX1. If this is the case, it should be demonstrated thoroughly.*

This is an excellent point. As the reviewer pointed out, p21 plays an important role in mediating the effects of p53 to induce G1 arrest after DNA damage. According to some papers, in the absence of p21, there is no G1 arrest after DNA damage (Waldman et al., Cancer Res, 1995). However, other papers (also cited in the Waldman et al. paper) show that p21 is not the only player (Deng et al., Cell, 1995).

*Can depletion of BTG2 and/or RRM2B and/or GPX1 recapitulate the effects of PINCR depletion?*

To address this point we have added new data (Figure 4—figure supplement 3 and Figure 4—figure supplement 4). We indeed found that depletion of *BTG2, GPX1* or *RRM2B* using siRNAs, recapitulates the effects of *PINCR* depletion in response to 5-FU treatment. Accordingly, we made changes in the text as described below.

“We next asked the question if depletion of the *PINCR* targets *BTG2, GPX1* and *RRM2B* recapitulated the effects of *PINCR* depletion. […] Significant reduction in G1 arrest after 5-FU treatment was observed after knockdown of *GPX1* but not *BTG2* or *RRM2B*. These data indicate that depletion of *BTG2, GPX1* or *RRM2B* recapitulates the effects of *PINCR* depletion in response to 5-FU treatment.”

Is it true that these genes are required for p21-dependent cell cycle arrest? This could be answered with siRNA knockdowns.

To address this question, we knocked down p21 alone or p21 and each of these *PINCR* targets and examined the effect on G1 arrest and apoptosis after 5-FU treatment. The results from 3 biological replicates (Figure 9) and the raw cell cycle profiles from a representative experiment (Figure 10) show that concurrent knockdown of p21 and BTG2 results in further reduction in G1 arrest and increased apoptosis as compared to knockdown of p21 alone. On the other hand, concurrent knockdown of p21 and RRM2B results in less G1 arrest and decreased apoptosis as compared to knockdown of p21 alone. These data indicate that in the absence of p21, BTG2 may function as a G1-arrest promoting gene in response to 5-FU, whereas RRM2B may function as a G1-arrest antagonizing gene upon 5-FU treatment. However, future investigations are needed to further understand the crosstalk between p21 and these genes during cell cycle arrest.

Author response image 1.(**A**) PINCR-WT HCT116 cells were transfected with CTL siRNA (siCTL) or p21 siRNAs (sip21) for 48 hr and p21 knockdown was measured by immunoblotting. GAPDH was used as loading control. (**B**) PINCRWT HCT116 cells were transfected for 48 hr with siCTL or sip21 or co-transfected with sip21 and siCTL or siBTG2 or siGPX1 or siRRM2B. PI-staining and FACS analysis was performed after 48 hr of 5-FU treatment. Error bars represent SD from three biological replicates. **p*<0.05; ^#^*p*<0.01; ***p*<0.005; ^##^*p*<0.001.**DOI:**
http://dx.doi.org/10.7554/eLife.23244.068

Author response image 2.Raw cell cycle profiles from a representative experiment for Figure 9.**DOI:**
http://dx.doi.org/10.7554/eLife.23244.069

*Is it true that p21 is unable to block CDKs, prevent RB phosphorylation and stall the cell cycle in the absence of PINCR? This should be explored in detail by looking a p21 localization, Rb phosphorylation and other markers of p21 action and G1-S progression.*

We have added new data (Figure 4—figure supplement 2) which shows that total p21 protein levels, p21 localization and Rb-phosphorylation are not regulated by PINCR. Accordingly, we made changes in the text, as described below.

“However, given the well-established role of p21 in controlling G1 arrest after DNA damage, it was important to make sure that loss of *PINCR* did not alter p21 protein levels, p21 subcellular localization and/or Rb- phosphorylation. […] These results indicate that p21 expression is not altered upon loss of *PINCR* and it can therefore be used as a negative control.”

*2) A gene-specific role for PINCR and Matrin. A key observation in this paper is that PINCR and Matrin are required for p53 transactivation at selective loci, and this is explained by a requirement of PINCR for p53 binding to the respective enhancers at those genes. The notion of a RNA (or a RNA binding protein) being required for p53 binding to some DNA elements but not others is very provocative and requires further investigation. Biochemically speaking, p53 is a potent DNA-binding protein that works as a tetramer to bind a 4-repeat DNA sequence (a 20 mer consensus made of two half sites with two palindromic repeats per half site).*

Our analysis of the overlap between the PINCR-regulated transcriptome identified by microarrays and the list of p53-regulated genes identified in the p53 GRO-seq paper (Allen et al., 2014) indicates that PINCR regulates the mRNA levels of only 11 p53 targets. This result is consistent with our additional data (Figure 1—figure supplement 5) showing that PINCR is expressed at ~25 molecules per HCT116 cell after DNA damage. A lncRNA expressed at ~25 molecules per cell is most likely not to regulate hundreds of p53 targets, unless it regulates p53 itself, which is not the case, as shown in Figure 4—figure supplement 1.

*How is this activity modified by RNA and/or a RNA-binding protein at some DNA elements but not others?*

This is an important question that was also raised by Reviewer # 3.

To address this point, we have added new data (Figure 7, Figure 7—figure supplement 1 and 2 and Figure 8). Briefly, we found that Matrin 3 associates with the enhancers of BTG2, GPX1 and RRM2B and this interaction is mediated by the binding of *PINCR* to Matrin 3. In addition, using published Hi-C data and ChIP- seq data for CTCF, RAD21, H3K4Me3, H3K27ac and p53, we provide evidence of chromatin looping between the enhancer region and p53RE in the promoter region of BTG2, GPX1 and RRM2B. As a control, Matrin 3 and PINCR do not associate with the p21 promoter. Accordingly, we made changes in the text as described below.

“Given the evidence that Matrin 3 associates with enhancer regions (Skowronska-Krawczyk et al., 2005), we reasoned that Matrin 3 modulates the induction of these genes by binding to their enhancer regions. […] These results indicate a role of Matrin 3 and *PINCR* in facilitating the association of p53 with the enhancers of specific p53 targets *BTG2, GPX1* and *RRM2B* and provide evidence of chromatin looping between the enhancers and promoters of these genes (Figure 7).”

Interestingly, when we looked at the sequence in the peaks of these enhancer regions, we were unable to find a canonical p53RE indicating indirect binding of p53.

“Finally, we examined the association of *PINCR-S1* with the p53RE and enhancer regions of *PINCR* targets in the presence and absence of p53 or Matrin 3. […] On the other hand, the observed loss in association of *PINCR-S1* to the enhancers and promoters upon knockdown of Matrin 3 indicates that Matrin 3 recruits *PINCR-S1* to these regions.”

*This is very tantalizing and hard to comprehend. In vitro protein-DNA binding experiments should be performed to investigate this. The genome-wide requirement for PINCR and Matrin should be defined. Is it true that only 11 p53 targets require PINCR?*

Our analysis of the overlap between the PINCR-regulated transcriptome identified by microarrays and the list of p53-regulated genes identified in the p53 GRO-seq paper (Allen et al., 2014) indicates that PINCR regulates the mRNA levels of only 11 p53 targets. This result is consistent with our additional data (Figure 1—figure supplement 5) showing that PINCR is expressed at ~25 molecules per HCT116 cell after DNA damage. A lncRNA expressed at ~25 molecules per cell is most likely not to regulate hundreds of p53 targets, unless it regulates p53 itself, which is not the case, as shown in Figure 4—figure supplement 1.

*If so, is it true that PINCR regulates p53 binding at only those 11 loci? Is it true that PINCR binds only to those p53 target loci? What about Matrin? What is Matrin's binding pattern relative to p53 target genes, those sensitive and insensitive to PINCR?*

One approach to address this question is by identifying genome-wide binding sites of p53, Matrin 3 and PINCR-S1. As suggested by the reviewers in the decision letter, we did not perform the Matrin 3 ChIP-seq or PINCR-S1 ChIP-seq experiments during the revision. However, our data (Figure 6 and Figure 8) shows that Matrin 3 and PINCR associate specifically with the p53RE of BTG2, RRM2B and GPX1 but not the p53RE of p21.

*These questions should be answered with careful ChIP-seq experiments for p53, PINCR-S1 and Matrin, and a detailed investigation of this 'gene-specific effects'. This is important, because the latest and most comprehensive investigations of p53 functionality at enhancers indicate that p53 does not employ cofactors to recognize and activate transcription from ~1000 enhancers in a wide variety of contexts (see for example Verfaillie et al., Genome Research 2016). This notion is further supported by earlier work demonstrating that p53 can recognize its p53REs in nucleosomal context, often leading to nucleosome displacement.*

As mentioned above, we did not conduct the ChIP-seq for Matrin 3 and PINCR- S1 during the revision. As requested by the reviewers, we performed p53 ChIP-seq in biological duplicates from PINCR-WT and PINCR-KO cells ± 5-FU. Unfortunately, in the first replicate there was too much background binding of p53 in one sample. In the second set, we observed specific p53 ChIP-seq peaks for only 41 among the 198 p53 targets identified in the p53 GRO-seq paper (Allen et al., 2016). Given this lack of significant overlap with well-established p53 target database and the fact that this data was from a one-time experiment, we believe that the p53 ChIP-seq experiments will need further optimization and we plan to investigate this in the future.

*3) DNA- versus RNA-dependent effects of PINCR. The authors conclude that PINCR acts by a RNA-dependent mechanism. However, this is based mostly on an unsatisfactory rescue overexpression experiment in PINCR knockout cells. This is important, because the field has been profoundly misled by earlier reports of another p53-inducible lncRNA, lincRNA-p21, which was first described to be acting by an RNA-dependent mechanism, yet now is ample clear that the lincRNA-p21 locus acts by a DNA-dependent, enhancer-like mechanism (see latest work from John Rinn's team in Cell Reports, backtracking on his original Cell paper by Huarte et al). To clarify this for PINCR, the authors should elaborate more on their knockout. What exact region of the genome did they delete when they deleted the 'entire PINCR region'? Did they delete the proximal p53 binding site as well? If they preserved the p53 enhancer, they should demonstrate that p53 binding to that region is intact by ChIP.*

We agree with the reviewer’s concern. To address the concern on the rescue experiment, we have added new data (Figure 4, Figure 4—figure supplement 5 and Figure 4—figure supplement 6, Figure 3—figure supplement 2–Figure 3—figure supplement 4) showing that PINCR knockdown using antisense oligos in HCT116 or knocking out PINCR in another CRC line SW48 cells, has phenotypes similar to PINCR-KO HCT116 cells. Accordingly, we have made changes in the text as described below:

“To make sure that the altered induction of *BTG2, GPX1* and *RRM2B* reflect a function of the *PINCR* transcript itself, we measured the induction of these genes after *PINCR* knockdown using antisense oligonucleotides (ASOs).[…] Taken together, the results from *PINCR* knockdown experiments corroborates our findings from the *PINCR*-KO clones.”

“Next, to determine if the observed phenotypes are not restricted to HCT116, we knocked out *PINCR* in SW48 cells (Figure 3—figure supplement 2). […] Moreover, following extended treatment with 5- FU, the *PINCR*-KO clone was markedly more sensitive than *PINCR*-WT clones (Figure 3—figure supplement 4). These data confirm that the phenotypic effects observed upon loss of *PINCR* are not unique to HCT116.”

To address the reviewer’s comment on the concern on the region of the genome that we deleted, we now show this deleted region (Figure 2—figure supplement 1). As shown in this figure, based on Sanger sequencing data, out of the 2 PINCR-KO clones used in our study, one clone (PINCR-KO#2) has intact p53RE whereas the other clone (PINCR-KO#1) has partially intact p53RE.

As suggested by the reviewer, we checked the binding of p53 to the p53RE of PINCR in both PINCR- KO clones and found that p53 binding was abrogated in the KO clone that had a partially intact p53RE.

*Other concerns:*

*1) Repeats and error bars in 2B.*

As suggested, we have repeated this experiment to generate error bars (Figure 1—figure supplement 2).

*2) Is PINCR upregulated in published GRO-seq data in HCT116 cells? It is unclear which ncRNA in Allen et al. the authors refer too. Is it refereed by a different name in Allen et al?*

PINCR was not identified in the published GRO-seq data in HCT116 cells. The reference Allen et al. in the original submission was cited to determine how many genes in our lncRNA arrays were also identified in the p53 GRO-seq paper.

*A genome viewer screenshot will go a long way to show that PINCR is truly a direct p53 target is truly direct.*

Genome viewer screenshot of p53 ChIP-seq data (Figure 1), the sequence of the p53RE (Figure 1—figure supplement 2) and our data (Figure 1) indicate that PINCR is a direct target of p53. In addition, as suggested by reviewer # 2, we have now deleted the p53RE in the PINCR promoter and performed luciferase assays. As shown in Figure 1, deletion of the p53RE resulted in marked reduction in luciferase activity under basal conditions and after DNA damage. Collectively, our data shows that PINCR is a direct target of p53.

*3) Raw cell cycle profiles for 3B-C.*

We have added these profiles (Figure 2—figure supplement 3). Please note that 3B-C in the original submission is Figure 2 in the revised manuscript.

*4) To confirm that PINCR is involved in p53-dependent G1 arrest, they should repeat the experiments they did with genotoxic agents using Nutlin instead.*

As suggested by the reviewer, we performed cell cycle analysis from PINCR-WT and PINCR-KO cells after Nutlin treatment. Results from this experiment are shown in Figure 2—figure supplement 5 and incorporated in the text, as described below.

“To confirm that *PINCR* is involved in p53-dependent G1 arrest, we performed cell cycle analysis from *PINCR*- WT and *PINCR*-KO cells after Nutlin-3 treatment. […] These data indicate that *PINCR* plays a role in p53-dependent G1 arrest and it has a prosurvival function in response to DNA damage”.

*Reviewer #2:*

*[…] 1) Since this is the first functional analysis of PINCR, additional basic characterization of the lncRNA is important. The authors should confirm the 5'/3' ends and splicing of the transcript by RACE and RT-PCR.*

This is a very important point. We have added new data (Figure 1—figure supplement 3 and Figure 1—figure supplement 4) to address this point. These results are described in the last paragraph of the subsection “PINCR is a nuclear lncRNA directly regulated by p53” and below.

As mentioned in the decision letter, we did not perform RACE. To identify the 5’ and 3’ends of the *PINCR* RNA, we analyzed our RNA-seq data from HCT116 cells (Li et al., unpublished). We found that the 5’and 3’end of the *PINCR* RNA matched the annotated transcript (Figure 1—figure supplement 3). Interestingly, the reads from the first exon were substantially higher than the reads from the last exon which may be due to 3’ to 5’ degradation of this lncRNA, as recently reported for other lncRNAs (Schlackow et al., 2017). Finally, to determine the length of the *PINCR* transcript, we performed RT-PCR using a forward primer at the start of exon 1 and reverse primer complementary to the last exon (exon 6). As a positive control, we used pCB6-*PINCR*, a mammalian expression construct in which the full-length annotated *PINCR* RNA (~2.2 kb) was inserted into pCB6. Because the reverse primer at the end of the annotated last exon did not work even for pCB6-*PINCR*, we had to use an alternative reverse primer that started at position 1864 of the annotated ~2.2 kb long *PINCR* RNA. As shown in (Figure 1—figure supplement 4), we found two closely migrating bands that matched the expected size of the amplicon (~1.8 kb).

*The copy number of the expressed transcript should be determined (e.g. +/- Nutlin, DOXO, and/or 5-FU).*

To estimate the number of molecules of *PINCR* per cell, we compared the FPKM of *PINCR* with the lncRNA *NORAD*, known to be expressed at 500-1000 molecules per HCT116 cell (Lee et al., 2016). This analysis indicated that *PINCR* is expressed at ~13-26 molecules per HCT116 cell after DNA damage and less than 1 molecule per cell without DNA damage (Figure 1—figure supplement 5). As an alternative approach, we performed qRT-PCR using in vitrotranscribed *PINCR* RNA and DOXO-treated HCT116 total RNA, and found that *PINCR* is expressed at ~27 molecules per HCT116 cell after DNA damage (Figure 1—figure supplement 5). These results are described in the second paragraph of page 6 of the revised manuscript.

*A northern blot would be helpful if possible to confirm the existence of a discrete transcript at the expected size.*

We tried the Northern blot from HCT116 (DOXO-treated) total RNA but it did not work. This is not surprising given our finding that PINCR is expressed at only ~25 molecules per HCT116 cell.

*The conservation pattern of the transcript and p53 response element should be commented upon.*

As suggested by the reviewer, we performed conservation analysis (Figure 1—figure supplement 6). We found that the *PINCR* promoter including the p53RE, mature *PINCR* transcript and the transcription start site are well conserved among primates but poorly conserved between human and mouse.

*2) The use of genome editing to stably delete PINCR is a good approach to establish a robust system for functional studies. However, the authors chose to delete a ~50 kb segment to remove the lncRNA which raises concerns regarding whether other important sequences were removed in addition to the 2.2 kb spliced PINCR sequence. I appreciate the authors' attempts to mitigate these concerns by performing a rescue experiment but the analysis of rescued cells does not go nearly far enough. Cell cycle analysis of rescued cells should be shown in addition to the provided analysis of cell survival after DOXO treatment (Figure 3). The clonogenic survival assays (Figure 3) should also be performed on rescued cells. Likewise, cell cycle, cell survival, and clonogenic survival of rescued cells after 5-FU treatment should be added to Figure 4.*

We have added the cell cycle analysis of the rescued cells. Reintroduction of PINCR in PINCR-KO cells significantly rescues the sub-G1 population but not the G1 arrest (Figure 2 and Figure 2—figure supplement 3). The incomplete rescue (rescue of cell death but not G1 arrest) may be because unlike endogenous *PINCR* that is induced ~100-fold after DOXO-treatment, the extent of exogenous *PINCR* overexpression was ~40-fold. Another possibility is that in the rescue experiments, we overexpressed the annotated isoform whereas we found that HCT116 cells express at least 2 isoforms of *PINCR* (Figure 1—figure supplement 4). Overall, the results from these rescue experiments indicate that the observed cell death phenotype after DOXO treatment of *PINCR*-KO cells was due to loss of *PINCR* RNA.

We would like to mention that in the interest of time, we did not perform the clonogenic survival assays and other suggested experiments with 5-FU. Instead, as shown below and as suggested in the decision letter, we addressed these issues using an orthogonal method in which we knocked down PINCR.

*Any concerns about the genome editing approach could be fully mitigated if the authors employed an orthogonal method to inhibit PINCR such as siRNA knockdown or CRISPRi, followed by analysis of cell cycle, survival, clonogenic growth, and target gene induction (e.g. BTG2, GPX1, RRM2B).*

This is an excellent point. As suggested by the reviewers in the decision letter, we have now examined the cell cycle, survival, clonogenic growth and target gene induction after knockdown of PINCR RNA. In collaboration with Ionis Pharmaceuticals, we tested 5 antisense oligos (ASOs) that target PINCR RNA. We achieved robust knockdown with only one ASO designated as PINCR-ASO. We have added the data from these experiments (Figure 4, Figure 4—figure supplement 5 and Figure 4—figure supplement 6) and made changes in the text as described below:

“To make sure that the altered induction of *BTG2, GPX1* and *RRM2B* reflect a function of the *PINCR* transcript itself, we measured the induction of these genes after *PINCR* knockdown using antisense oligonucleotides (ASOs). […] Taken together, the results from *PINCR* knockdown experiments corroborates our findings from the *PINCR*-KO clones.”

*3) While p53-mediated PINCR induction was demonstrated in multiple cell lines, all functional analyses are limited to HCT116. PINCR should be knocked out or inhibited in a second cell line to confirm that the phenotypic effects observed are not unique to HCT116.*

This is an important point. As suggested by the reviewer, we knocked out PINCR in another CRC line (SW48 cells) to confirm that the phenotypic effects observed are not unique to HCT116. We found that loss of PINCR in SW48 cells results in reduced G1 arrest, increased apoptosis and reduced clonogenicity in response to 5-FU. Accordingly, we made changes in the text as described below:

“Next, to determine if the observed phenotypes are not restricted to HCT116, we knocked out *PINCR* in SW48 cells (Figure 3—figure supplement 2). […] These data confirm that the phenotypic effects observed upon loss of *PINCR* are not unique to HCT116.”

*4) Knockdown of Matrin 3 partially phenocopies PINCR knockout with respect to cell survival after 5-FU treatment (Figure 6). As this is a very important prediction of their model, the analysis of Matrin 3 knockdown should be extended. First, at least 2 independent siRNAs should be tested to avoid off-target effects (this is especially important when monitoring general cellular phenotypes such as survival). Furthermore, the cell cycle should be examined after knockdown of Matrin 3 and 5-FU treatment and both cell cycle and apoptosis should be examined after DOXO treatment.*

We agree with the reviewer’s comment. As suggested, we now provide evidence with 2 independent Matrin siRNAs, that knockdown of Matrin 3 (Figure 5 and Figure 5—figure supplement 1) results in increased cell death in PINCR-WT cells but not in PINCR-KO cells after 5-FU treatment (Figure 5). However, the effect on G1 arrest was consistently observed (Figure 5—figure supplement 2) with only one siRNA (siMatrin 3-I). Cell cycle analysis in response to DOXO-treatment showed that knockdown of Matrin 3 resulted in increased sub-G1 population (Figure 5—figure supplement 2) but the effect on G1 arrest was not the same for both siRNAs (Figure 5—figure supplement 2). Raw cell cycle profiles from a representative experiment are shown in Figure 5—figure supplement 3.

These data suggest that Matrin 3 mediates the anti-apoptotic effect of PINCR in response to DNA damage induced by 5-FU or DOXO. Furthermore, in *PINCR*-WT but not in PINCR-KO cells, silencing Matrin 3 resulted in less or no induction of the *PINCR* targets *BTG2, GPX1* and *RRM2B* (Figure 5) after 5-FU treatment.

*5) What happens to the Matrin-p53 interaction in PINCR KO cells? Co-IP assays (as in Figure 7) should be performed in these cells.*

As suggested by the reviewer, we performed the p53-Matrin 3 Co-IP experiment. We found no difference in the amount of p53 pulled down in the Matrin 3 IP material between PINCR-WT and PINCR-KO cells (Figure 11). This result is consistent with our data (Figure 6) where we found that the p53-Matrin 3 interaction is RNA-independent.

Author response image 3.PINCR-WT and PINCR-KO HCT116 cells were treated with 5-FU for 48 hr and the interaction between p53 and Matrin 3 was assessed by immunoblotting for p53 following IgG or Matrin 3 IP.**DOI:**
http://dx.doi.org/10.7554/eLife.23244.070

*Likewise, does PINCR associate with p53 response elements in the absence of Matrin and/or p53? The experiments in Figure 7 should be repeated after Matrin and p53 knockdown.*

Great suggestion. As suggested, we examined the association of *PINCR-S1* with the p53RE and enhancer regions of *PINCR* targets in the presence and absence of p53 or Matrin 3. To do this, we knocked down p53 (Figure 8—figure supplement 5) or Matrin 3 with siRNAs and performed streptavidin pulldowns. As compared to the p53RE in the *PINCR* target gene promoters, in CTL siRNA transfected cells treated with 5-FU, we found stronger association of *PINCR-S1* with the enhancers of the *PINCR* targets (Figure 8). Silencing Matrin 3 or p53 resulted in dramatic reduction in the association of *PINCR-S1* with these enhancers and p53REs (Figure 8). Because p53 is important for *PINCR* expression, it is likely that the reduced association of *PINCR-S1* to these regions after p53 knockdown is due to lack of *PINCR* expression. On the other hand, the observed loss in association of *PINCR-S1* to the enhancers and promoters upon knockdown of Matrin 3 indicates that Matrin 3 recruits *PINCR-S1* to these regions.

*6) While the observation that p53-mediated induction of BTG2, GPX1, and RRM2B is attenuated in PINCR KO cells and after Matrin 3 knockdown is interesting, there is no evidence that these changes in gene expression are responsible for the impaired p53-mediated cell cycle arrest and apoptosis in PINCR KO cells. Does knockdown of any of these targets phenocopy PINCR KO?*

This point was also raised by reviewer #1. To address this point we have added new data (Figure 4—figure supplement 3 and Figure 4—figure supplement 4). We indeed found that depletion of *BTG2, GPX1* or *RRM2B* using siRNAs, recapitulates the effects of *PINCR* depletion in response to 5-FU treatment. Accordingly, we made changes in the text, as described below.

“We next asked the question if depletion of the *PINCR* targets *BTG2, GPX1* and *RRM2B* recapitulated the effects of *PINCR* depletion. […] These data indicate that depletion of *BTG2, GPX1* or *RRM2B* recapitulates the effects of *PINCR* depletion in response to 5-FU treatment.”

*Does overexpression of any of these genes rescue PINCR KO?*

As mentioned in our initial response, we could have attempted this experiment but in the interest of time and given that overexpression of some of these genes in the absence of DNA damage is known to induce cell cycle arrest, we were unable to perform this experiment during the revision of our study.

*7) The p53-RE luciferase assays in Figure 2 would be much more convincing if a reporter with mutations in the p53-RE were tested in parallel as a negative control.*

As suggested, in the revised manuscript, we performed luciferase assays after deleting the p53RE in the luciferase reporter carrying the PINCR promoter. As shown in Figure 1, we found that deletion of the p53RE in the PINCR promoter resulted is profound reduction in luciferase activity, both in the absence and presence of DOXO.

*8) In Figure 7—figure supplement 1, it appears that the polyadenylation signal was removed when the S1 tag was inserted (at the end of the red sequence). Does this affect the level of PINCR expression? Although the authors show PINCR-S1 induction after DOXO treatment, PINCR-S1 levels should be directly compared to endogenous PINCR.*

This is a very interesting point. The reviewer is correct. The potential polyadenylation signal was removed when the S1 tag was inserted. We call this as a potential polyadenylation signal because according the RefSeq annotation, PINCR is not a polyadenylated lncRNA but this needs to be confirmed in future experiments. As suggested by the reviewer, we compared PINCR-S1 levels to endogenous PINCR by qRT-PCR (normalized to GAPDH) in PINCR-S1 HCT116 cells and in parental HCT116 cells. We found that PINCR-S1 is expressed at levels similar to *PINCR* (Figure 8—figure supplement 3). These data indicate that insertion of the S1-tag does not affect PINCR expression.

*Reviewer #3:*

*[…] 1) Figure 4. In view of the suggested biology of PINCR, one might expect it to be*

*selectively LESS induced under conditions that favor p53-dependent cell death. What happens, for example, upon treatment of HCT116 cells with high dose 5-FU? As a matter of fact, the data in Figure 4 and its supplemental already point in this direction: 5-FU, at the concentration used by the authors, induces more apoptosis (11%) than the other treatments, while causing a relatively modest increase in PINCR. Thus, the relative extent of endogenous PINCR induction might be a determinant of the likelihood of the cells to die rather than arrest. I suggest that the authors address this possibility more closely, e.g. by employing several different concentrations of 5-FU and comparing the extent of PINCR induction, induction of a typical pro-apoptotic p53 target gene (e.g. PUMA or CD95), and the accrual of sub-G1 cells.*

This is a very interesting point. As suggested, we treated the cells with several different concentrations of 5-FU and measured the induction of PINCR and PUMA. As shown in Figure 3—figure supplement 5, both PINCR and PUMA are induced ~10-14-fold after treatment of PINCR-WT cells with 0, 10, 50, 100 or 375 μm of 5-FU. The sub-G1 cells for this range of 5-FU was 0, 3, 8, 14 and 43% in PINCR-WT cells and 0, 9, 17, 34 and 83% in PINCR-KO cells (Figure 3—figure supplement 5). In addition, at each dose of 5-FU, the% of cells arrested in G1 was smaller in PINCR-KO cells as compared to PINCR-WT cells (Figure 3—figure supplement 5). Based on these data, it appears that the relative extent of endogenous PINCR induction may not be a determinant of the likelihood of the cells to die rather than arrest.

*2) Figure 4. The in vivo result is impressive. However, it does not necessarily support a pro-survival function of PINCR as concluded in the subsection “PINCR loss results in hypersensitivity to 5-FU and decreased tumor growth”: the authors did not address survival/death at all in this experiment. It will be important to perform histopathological examination of the tumors, and show (by cleaved caspase 3 or TUNEL) that the PINCR KO tumors indeed manifest a greater extent of apoptosis.*

This is an important point. As suggested, we performed immunohistochemical staining of the tumors for the proliferation marker Ki67 and the apoptosis marker cleaved caspase-3 (Figure 3—figure supplement 7). We found that both *PINCR*-WT and *PINCR*-KO tumors had a high proportion of Ki67- positive cells (>50%) and a very low proportion of cleaved caspase-3-positive cells (<1%). As compared to *PINCR*-WT tumors, the *PINCR*-KO tumors had significantly decreased Ki67-positive cells (Figure 3—figure supplement 7), suggesting that the observed reduced tumor volume is due to inhibition of cell proliferation.

Based on these data, we have accordingly modified the text on page 11. We agree that the in vivo data does not support a pro-survival function of PINCR. However, our in vitro data from 2 CRC cell lines suggests that PINCR has a prosurvival function in response to DNA damage.

*3) I could not find any ChIP experiment showing that matrin 3 is required for optimal binding of p53 to the p53RE's of BTG2 etc. Did I miss it? This is an important experiment, as the authors attribute the reduced expression of those genes to reduced recruitment of p53 to their p53RE's.*

We agree with the reviewer. In the revised manuscript, we show that the binding of p53 to the p53RE element of BTG2, GPX1 and RRM2B is not affected after knockdown of Matrin 3 (Figure 6 and Figure 6—figure supplement 2). This was a surprising result. However, our analysis of the enhancer region of these genes and potential chromatin looping between their promoter and enhancer regions indicates that Matrin 3 may regulate these genes not by controlling p53 binding at their promoters but by modulating their expression after 5-FU treatment by associating with their enhancer regions (Figure 7 and Figure 7—figure supplement 1 and Figure 7—figure supplement 2).

*4) Why does PINCR bind only to a subset of p53RE's? Is there significant homology between the sequence of PINCR and that of the DNA in the vicinity of those p53RE's, but not the p21 p53RE for example?*

This is a very interesting question. We did not find significant homology between the PINCR RNA and the DNA in the vicinity of the p53RE of BTG2, GPX1 and RRM2B. Our data indicates that PINCR binds to Matrin 3 and after binding to Matrin 3, PINCR is recruited to the p53RE of these genes but not the p21 p53RE as shown in Figure 6 and Figure 6—figure supplement 1. This is further supported by our data showing PINCR does not associate to these p53REs upon Matrin 3 knockdown (Figure 8).

*5) Figure 3: How does PINCR overexpression affect the cell cycle profile and the expression of the specific subset of G1-related genes without and with DNA damage? Is PINCR overexpression alone sufficient to upregulate those genes?*

Another great suggestion. We found that PINCR overexpression has no effect on the expression of BTG2, GPX1 and RRM2B, without and with DNA damage induced by 5-FU (Figure 8—figure supplement 4). In addition, there was no effect on cell cycle profile after PINCR overexpression (Figure 3—figure supplement 6). Thus, our data indicates that PINCR overexpression alone is not sufficient to upregulate those genes. PINCR functions under conditions that upregulate p53, such as Nutlin treatment or DNA damage induced with DOXO or 5-FU.

*6) Figure 7: What happens to matrin 3 after DNA damage? Do its levels increase? Does it become more localized to the nucleus?*

As suggested, we measured total Matrin 3 protein levels and nuclear and cytoplasmic Matrin 3 protein levels in HCT116 cells, in untreated condition and after DNA damage induced by 5-FU. We found no change in total, nuclear or cytoplasmic Matrin 3 levels in response to DNA damage (Figure 5—figure supplement 5).

*7) Does PINCR regulate its own expression after DNA damage?*

To determine if PINCR regulates its own expression after DNA damage, we introduced exogenous PINCR (using pCB6-PINCR) in PINCR-S1 cells and treated the cells with 5-FU. Using PINCR-S1-specific primers, we found that PINCR overexpression had no effect on PINCR-S1 levels (Figure 8—figure supplement 4).

*Does PINCR knockout affect the binding of p53 to the PINCR promoter?*

Based on Sanger sequencing data, we found that out of the 2 PINCR-KO clones used in our study, one clone (PINCR-KO#2) has intact p53RE whereas the other clone (PINCR-KO#1) has partially intact p53RE. The DNA sequence showing the exact region of the genome that was deleted is shown in Figure 2—figure supplement 1. As suggested, we checked the binding of p53 to the p53RE of PINCR in both PINCR-KO clones and found that p53 binding was abrogated in the KO clone that had a partially intact p53RE (Figure 2—figure supplement 2).

*8) Figure 7. Does matrin 3 bind those promoters also without DNA damage, or is it recruited only after DNA damage?*

It depends on the gene. As shown in Figure 6, in untreated cells, Matrin 3 binds to the p53RE of BTG2 and RRM2B but not GPX1. The binding of Matrin 3 to all these 3 p53REs is increased after DNA damage (Figure 6).

*9) Does matrin 3 affect the induction of BTG2 etc. in response to Nutlin, or is this DNA damage-specific?*

Our new data (Figure 5—figure supplement 4) shows that Matrin 3 regulates the induction of PINCR targets in response to Nutlin treatment as well.